# Physiological Signal Embeddings (PHASE) via Interpretable Stacked Models

## Abstract

In health, machine learning is increasingly common, yet neural network embedding (representation) learning is arguably under-utilized for physiological signals. This inadequacy stands out in stark contrast to more traditional computer science domains, such as computer vision (CV), and natural language processing (NLP). For physiological signals, learning feature embeddings is a natural solution to data insufficiency caused by patient privacy concerns – rather than share data, researchers may share informative *embedding models* (i.e., representation models), which map patient data to an output embedding. Here, we present the PHASE (PHysiologicAl Signal Embeddings) framework, which consists of three components: i) *learning neural network embeddings* of physiological signals, ii) *predicting outcomes* based on the learned embedding, and iii) *interpreting the prediction results* by estimating feature attributions in the "stacked" models (i.e., feature embedding model followed by prediction model). PHASE is novel in three ways: 1) To our knowledge, PHASE is the first instance of transferal of neural networks to create physiological signal embeddings. 2) We present a tractable method to obtain feature attributions through stacked models. We prove that our stacked model attributions can approximate Shapley values – attributions known to have desirable properties – for arbitrary sets of models. 3) PHASE was extensively tested in a cross-hospital setting including publicly available data. In our experiments, we show that PHASE *significantly outperforms* alternative embeddings – such as raw, exponential moving average/variance, and autoencoder – currently in use. Furthermore, we provide evidence that *transferring neural network embedding/representation learners* between distinct hospitals can yield performant embeddings and offer recommendations when transference is ineffective.

## 1 Introduction

Representation learning (i.e., learning embeddings) (Bengio et al., 2013) has been applied to medical images and clinical text (Tajbakhsh et al., 2016; Ravishankar et al., 2016; Lv et al., 2014) but has been under-explored for time series physiological signals in electronic health records. This paper introduces the PHASE (PHysiologicAl Signal Embeddings) framework to learn embeddings of physiological signals (Figure 1a), which can be used for various prediction tasks (Figure 1b), and has been extensively tested in terms of its transferability using data from multiple hospitals (Figure 1d). In addition, this paper introduces an interpretability method to compute per-sample feature attributions of the original features (i.e., not embeddings) for a prediction result in a tricky "stacked" model situation (i.e., embedding model followed by prediction model) (Figure 1c).

Based on computer vision (CV) and natural language processing (NLP), exemplars of representation learning, physiological signals are well suited to embeddings. In particular, CV and NLP share two notable traits with physiological signals. The first is *consistency*. For CV, the domain has consistent features: edges, colors, and other visual attributes. For NLP, the domain is a particular language with semantic relationships consistent across bodies of text. For sequential signals, physiological patterns are arguably consistent across individuals. The second attribute is *complexity*. Across these three domains, each particular domain is sufficiently complex such that learning embeddings is non-trivial. Together, consistency and complexity suggest that for a particular domain, every research group independently spends a significant time to learn embeddings that may ultimately be

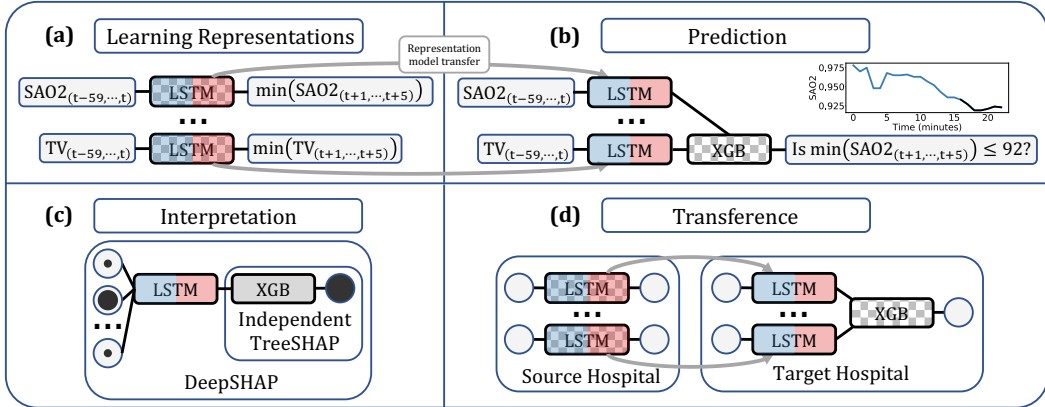

Figure 1: The PHASE framework, which consists of embedding learning, prediction, interpretation, and transference. The checkered patterns denote that a model is being trained in the corresponding stage, whereas solid colors denote fixed weights/models. The red side of the LSTM denotes the hidden layer we will use to generate embeddings. In (c), the size of the black circles on the left represent the feature attributions being assigned to the original input features. The signals and the outputs of the LSTMs are vectors. Multiple connections into a single XGB model are simply concatenated. More details on the experimental setup can be found in Sections 4.1 and 6.1.

quite similar. In order to avoid this negative externality, NLP and CV have made great progress on standardizing their embeddings; in health, physiological signals are a natural next step.

Furthermore, physiological signals have unique properties that make them arguably better suited to representation learning than traditional CV and NLP applications. First, physiological signals are typically generated in the health domain, which is constrained by patient *privacy concerns*. These concerns make sharing data between hospitals next to impossible; however, sharing models between hospitals is intuitively safer and generally accepted. Second, a key component to successful transfer learning is a *community* of researchers that work on related problems. According to Faust et al. (2018), there were at least fifty-three research publications using deep learning methods for physiological signals in the past ten years. Additionally, we discuss particular examples of neural networks for physiological signals in Section 2.2. These varied applications of neural networks imply that there is a large community of machine learning research scientists working on physiological signals, a community that could one day work collaboratively to help patients by sharing models.

Although embedding learning has many aforementioned advantages, it makes interpretation more difficult. Naive applications of existing interpretation methods (Shrikumar et al., 2016; Sundararajan et al., 2017; Lundberg & Lee, 2017; Lundberg et al., 2018) do not work for models trained using learned embeddings, because they will assign attributions to the embeddings. Feature attributions assigned to embeddings will be meaningless, because the embeddings do not map to any particular input feature. Instead, each embedding is a complicated, potentially non-linear combination of the original raw physiological signals. In a health domain, inability to meaningfully interpret your model is unsatisfactory. Healthcare providers and patients alike generally want to know the reasoning behind predictions/diagnoses. Interpretability can enhance both scientific discovery as well as provide credibility to predictive models. In order to provide a principled methodology for mapping embedding attributions back into physiological signal attributions, we provide a proof that justifies PHASE's Shapley value framework in Section 3.3. This framework generalizes across arbitrary stacked models and currently encompasses neural network models (e.g., linear models, neural networks) and tree-based models (e.g., gradient boosting machines and random forests).

In the following sections, we discuss previous related work (Section 2) and describe the PHASE framework (Section 3). In Section 4, we first evaluate how well our neural network embeddings make accurate predictions (Section 4.2.1). Second, we evaluate whether transferring these embedding learners still enables accurate predictions across three different hospitals separated by location and across hospital departments (Section 4.2.2). Lastly, we present a visualization of our methodology for providing Shapley value feature attributions through stacked models in Section 4.2.3.

## 2 RELATED WORK

### 2.1 REPRESENTATION LEARNING IN THE HEALTH DOMAIN

Representation learning (embedding learning) in health is growing more popular. One particularly natural subdomain is medical image analysis, e.g., mammography analysis, kidney detection in ultrasound images, optical coherence tomography image analysis, diagnosing pneumonia using chest X-ray images, lung pattern analysis, otitis media image analysis, and more (Arevalo et al., 2016; Ravishankar et al., 2016; Kermany et al., 2018; Liao et al., 2013; Christodoulidis et al., 2016; Shie et al., 2015). Outside of image analysis, additional examples of transfer learning in the medical domain include Lv et al. (2014), Wiens et al. (2014), Brisimi et al. (2018), Choi et al. (2017), Choi et al. (2016), and Che et al. (2016). Even within physiological signals, some examples of embedding learning are beginning to sprout up, including Wu et al. (2013), who utilize kNNs to perform transfer learning for brain-computer interaction. Comparatively, PHASE transfers neural networks as embedding functions learned in an partially supervised manner, where the embeddings provide a basis for training a model on any prediction task (as opposed to being tied to the prediction they were trained on). We denote partially supervised networks to be networks trained with prediction tasks related to the final downstream prediction.

### 2.2 NEURAL NETWORKS FOR PHYSIOLOGICAL SIGNALS

To our knowledge, our work is the first to transfer deep neural networks for embedding sequential physiological signals, embeddings which are not tied to a particular prediction problem. One caveat is that supervised deep learning can be said to inherently learn embeddings. In physiological signals, there are several examples of particular supervised learning tasks with neural networks. Srinivasan et al. (2007) and Guo et al. (2010) both detect epilepsy from signals, Wilson & Russell (2003) utilize psycho-physiological measurements to assess mental workload, Wagner et al. (2005) and Chanel et al. (2006) utilize physiological signals to classify emotions, Koike & Kawato (1995) reconstruct human arm movement from EMG signals, Sullivan et al. (2010) reconstruct missing physiological signals, and Yang & Hsieh (2016) use acoustic signals to detect anomalies in heart sound. Based on this substantive community of research scientists working on physiological signals, there is a clear opportunity to unify independent research by appropriately using *partially supervised* feature embedding learning.

In the vein of embedding learning, Martinez et al. (2013) applied autoencoders to blood volume pulse and skin conductance measured from 36 people playing a video game and used the encodings to predict affective state. In their paper, the sample size is fairly small and reflects that their primary objective was to perform feature extraction and feature selection. In contrast, PHASE evaluates transferring embedding learners (i.e., feature extractors) across multiple hospitals (Table 1).

### 2.3 FORECASTING FOR OPERATING ROOM DATA

Lundberg et al. (2017) proposed an approach, namely Prescience, which achieved state-of-the-art hypoxemia predictions using operating room data, the same data we used to evaluate PHASE. Prescience utilizes gradient boosting machines (GBM) applied to features extracted by using traditional time series feature extraction methods – exponential moving average/variance embeddings. Prescience compares prediction models including gradient boosting machines, linear lasso, linear SVM, and a parzen window with the objective of forecasting low blood oxygen in the future. Ultimately, Lundberg et al. (2017) find the highest performing method to be GBM trees. With samples drawn from the same data set, PHASE seeks to substitute the feature extraction used in Prescience with a deep learning approach, which resulted in a better average precision compared to Prescience without the clinical text features (4 is Prescience and 12 is PHASE in Figure 2).

### 2.4 FEATURE ATTRIBUTIONS FOR INTERPRETABILITY

Interpretability of models via local feature attributions has been addressed in numerous independent pieces of work (Shrikumar et al., 2016; Sundararajan et al., 2017; Lundberg & Lee, 2017; Lundberg et al., 2018). For our evaluation of interpretability, we choose to focus on Shapley values introduced by Lloyd Shapley, originally in the context of game theory (Shapley, 1953). Lundberg & Lee (2017)

identify Shapley values as the only additive feature attribution method that satisfies the properties of local accuracy, missingness, and consistency. For PHASE, our pipeline includes multiple models – GBM trees and LSTM networks. Methods exist for obtaining Shapley values for GBM trees (Tree SHAP) and for neural networks (Deep LIFT/Deep SHAP) (Lundberg et al., 2018; Shrikumar et al., 2016; Lundberg, 2018). However, the default version of these methodologies do not beget a theoretically justified approach for propagating attributions through multiple models. In fact, to the authors' knowledge, a tractable method for obtaining local feature attributions for mixes of neural networks and trees does not exist. In this paper, we utilize versions of Tree SHAP and Deep LIFT that create single reference attributions that can be composed to address stacked models. At the end of obtaining many attributions, the average of these attributions approximates the Shapley value attributions (more details in Section 3.3).

## 3 OUR APPROACH: PHYSIOLOGICAL SIGNAL EMBEDDINGS (PHASE)

Taking inspiration from sinusoidal waveforms, we name our methodology PHASE. In the PHASE framework, the first step is to learn neural network embeddings for physiological signals (Figure 1a). The second step is to predict outcomes based on the learned feature embedding (as in Figure 1b), potentially across multiple hospitals (as in Figure 1d). Finally, the last step is to interpret the prediction results by estimating feature attributions through the models trained in the first two steps (as in Figure 1c).

### 3.1 LEARNING EMBEDDINGS - LONG SHORT TERM MEMORY (LSTM) NETWORKS

PHASE uses LSTM networks to learn feature embeddings from time series physiological data. LSTMs are a popular variant on recurrent neural networks introduced by Hochreiter & Schmidhuber (1997). They have the capacity to model long term dependencies, while avoiding the vanishing gradient problem (Pascanu et al., 2012). Details on the model architecture may be found in Section 6.2.

For PHASE, we first train a univariate LSTM on each physiological signal $\mathcal{P}$, predicting the minimum of $\mathcal{P}$ in the future five minutes (Figure 1a). Note that we choose the minimum of the next five minutes because we care about forecasting adverse outcomes. Then, we obtain hidden embeddings of the original physiological signals by passing them through to the hidden layer (the red layer in Figure 1a). These embeddings are unsupervised in the sense that training them simply requires the same feature (albeit at different time steps). Yet, they are supervised in that we specify our interest to be forecasting adverse outcomes constituted by low signal values. We choose to focus on the minimum, because adverse outcomes in physiological signals are often tied to too-low signals. We find that the completely unsupervised alternative of an LSTM autoencoder is significantly less performant than the LSTM trained to predict the minimum of the next five minutes (8 is autoencoder and 12 is PHASE in Figure 2).

One reason behind having *univariate* neural networks is for transference. By using univariate networks, the input to the final prediction model may be any set of physiological signals with existing embedding learners. This is especially useful because hospital departments have substantial variation in the signals they may choose to collect for features. Another reason for univariate networks is that data in a single hospital is often collected at different points in time, or new measurement devices may be introduced to data collection systems. For traditional pipelines, it may be necessary to re-train entire machine learning pipelines when new features are introduced. With univariate networks, the flexibility would mean pre-existing embedding learners would not necessarily need to be re-trained.

### 3.2 PREDICTION - GRADIENT BOOSTING MACHINES (GBM)

PHASE can use any prediction model. In this paper, we focus on gradient boosting machine trees because the Prescience method found that they outperform several other models in the operating room data (Lundberg et al., 2017). Gradient boosting machines were introduced by Friedman (2001). This technique creates an ensemble of weak prediction models in order to perform classification/regression tasks in an iterative fashion. In particular, we utilize XGBoost, a popular implementation of gradient boosting machines that uses additive regression trees (Chen & Guestrin,

2016). XGBoost often dominates in Kaggle, a platform for predictive modeling competitions. In particular, seventeen out of twenty nine challenge winning solutions used XGBoost in 2015 (Chen & Guestrin, 2016). For PHASE, we postulate that utilizing embeddings of time series signals provides stronger features for the ultimate prediction with XGB (as visualized in Figure 1b). Details on the model architecture may be found in Section 6.2.

## 3.3 INTERPRETATION - SHAPLEY VALUES THROUGH STACKED MODELS

PHASE addresses an inherent challenge in the *interpretation* of an embedding model (or feature representation model). Estimating feature attributions is a common way to make a prediction result interpretable. At a high level, the goal is to explain how much each feature matters for a model's prediction. However, this goal is only meaningful if the model being explained uses features with a natural human interpretation. For example, if we interpret PHASE's GBM model, which takes the embeddings as input and outputs a prediction, our feature attributions will be assigned to the embeddings, which are not meaningful to doctors or patients.

The answer is to extend the prediction model (here, a GBM) by combining the feature embedding model (here, a LSTM network), which makes a "stacked" model (Figure 1c). Since the original features in the embedding stage are meaningful, one solution is to utilize a model agnostic feature attribution method over the "stacked" model. For our attributions, we aim to provide Shapley values, for which the exact model agnostic computation has an exponential computational complexity ($O(N2^M)$, where $N$ is the sample size and $M$ is the number of features) (Shapley, 1953). In response, one might want to use a model-specific method of computing approximate Shapley values to gain speed by knowing the model. However, to the authors' knowledge, there was previously no model-specific method to estimate Shapley values for a stack comprised of LSTMs and a GBM (or even local feature attributions for that matter).

**Single reference Shapley values:** Our new method for estimating Shapley values for the aforementioned stacked model (i.e., LSTMs and GBM), requires adaptations on two existing feature attributions methods. First is Deep SHAP, a variant on Deep LIFT – a feature attribution method for neural networks (Lundberg, 2018; Shrikumar et al., 2016). Deep SHAP differs from Deep LIFT in that it can find attributions for single references. Both methods can be written as modifications to a traditional backward pass through a neural network (Ancona et al., 2018). Since the computational complexity of a backward pass is the same as a forward pass through the network, we can consider this cost "low". The second method we utilize is "Independent Tree SHAP". This method is a variation on normal Tree SHAP (Lundberg et al., 2018), but it can be computed for single references. Independent Tree SHAP has a computational complexity of $O(MLT)$, where $L$ is the maximum number of leaves in any given tree and $T$ is the number of trees in the GBM.

**"Stacked" model Shapley values:** Combining these two methods amounts to treating the "stacked" model (Figure 1c) as a larger neural network and applying Deep SHAP to pass back attributions as gradients at each layer (Ancona et al., 2018). However, at the GBM layer we obtain the appropriate gradients by dividing the Independent Tree SHAP Shapley values by the difference between the sample and the references. According to Theorem 1, we can then average over these single reference attributions for an approximation to the Shapley values.

**Generalizability:** Note that Theorem 1 also implies that for any arbitrary set of models in a stack, if single reference Shapley values are obtainable for each model, the Shapley values for the entire stack can be obtained. Because the single reference Shapley value methods are known for neural networks and for trees, any "stacked" model composed of these two methods can be explained. Worth noting is that many embedding/prediction models can be represented as neural networks, making our framework to attribute "stacked" models fairly general.

**Theorem 1.** *Computing the average over single reference Shapley values approaches the true Shapley values.*

*Proof.* Starting with the definition of Shapley values:

$$\phi_i = \sum_{S \subseteq F \setminus \{i\}} \frac{|S|!(|F| - |S| - 1)!}{|F|!} \big(f_{S \cup \{i\}}(x) - f_S(x)\big)$$

$$= \sum_{S \subseteq F \setminus \{i\}} \frac{|S|!(|F| - |S| - 1)!}{|F|!} \big(\mathbb{E}_{\mathcal{D}}[f(x)|x_{S \cup \{i\}}] - \mathbb{E}_{\mathcal{D}}[f(x)|x_S]\big)$$

where $\mathcal{D}$ is the data distribution, $F$ is the set of all features, and $f$ is our model. Rewriting the sum over all permutations of $F$, rather than over all combinations, the weighting term becomes one:

$$\phi_i = \sum_{S_p \subseteq F \setminus \{i\}} \mathbb{E}_{\mathcal{D}}[f(x)|x_{S_p \cup \{i\}}] - \mathbb{E}_{\mathcal{D}}[f(x)|x_{S_p}]$$

$$= \mathbb{E}_F[\mathbb{E}_{\mathcal{D}}[f(x)|x_{S_p \cup \{i\}}] - \mathbb{E}_{\mathcal{D}}[f(x)|x_{S_p}]]$$

$$= \mathbb{E}_{\mathcal{D}}[\mathbb{E}_F[f(x)|x_{S_p \cup \{i\}}] - \mathbb{E}_{N_p}[f(x)|x_{S_p}]]$$

where the last step depends on independence between the permutations, independence between the conditional and non-conditional sets, and the data generating mechanism. $\square$

## 4 EXPERIMENTS

We first describe our data sets, evaluation metric, model architectures, and the results of comparisons between PHASE and alternative approaches in various testing scenarios (Section 4.2). More details on the model architecture and the models used for each experiment can be found in the Appendix.

### 4.1 EXPERIMENTAL SETUP

**Data Description** Hospital 0/1 data was collected via the Anesthesia Information Management System (AIMS), which records all data measured in the operating room during surgery. Both medical centers are within the same city (within 10 miles of each other). Hospital P is a sub-sampled version of the publicly available MIMIC data set from PhysioNet, which contains data obtained from an intensive care unit in Boston, Massachusetts (Johnson et al., 2016). Hospital P data was collected several thousands of miles from the medical centers associated with hospital 0/1 data. Some details about these hospitals are in Table 1, with more in Appendix: Section 6.1.

| **Model** | Department | # Procedures | # Hypoxemia Samples | Base Rate |
|---|---|---|---|---|
| Hospital 0 | OR | 29,035 | 3,528,507 | 1.09% |
| Hospital 1 | OR | 28,136 | 3,751,163 | 2.18% |
| Hospital P | ICU | 1,669 | 5,080,864 | 3.93% |

Table 1: Statistics of the different data sources. Hospital P is a public data set (PhysioNet). Additional details about hospital 0/1 data in Figures 5, 6, and 7 in Appendix: Section 6.1

The hospital 0/1 data includes static information (height, weight, age, sex, procedure codes), as well as real-time measurements of thirty-five physiological signals (e.g., $SaO_2$, $FiO_2$, $ETCO_2$, etc.) sampled minute by minute. Although the hospital P data contains several physiological signals sampled at a high frequency, we solely use a minute by minute $SaO_2$ signal for our experiments. Any missing values in the data are imputed by the mean and each feature is standardized to have unit mean and variance. One important note is that although hospitals 0 and 1 are spatially close, one is an academic medical center and one is a trauma center. More details on the patients from these hospitals can be found in Section 6.1.

**Evaluation Methodology** PHASE and alternative approaches are evaluated based on real-time prediction tasks, for example, whether a certain condition will occur in the next 5 minutes (details in Sections 4.2 and 6.1). Our evaluation metric for prediction performance in binary classification is area under the precision-recall curve, otherwise known as *average precision* (AP). Rather than ROC curves, PR curves often better highlight imbalanced labels. Precision is defined as $\frac{tp}{tp+fp}$ and recall is $\frac{tp}{tp+fn}$, where $tp$ is true positives, $fp$ false positives, and $fn$ false negatives. The area under the curve provides a summary statistic that balances both precision and recall.

## 4.2 RESULTS

In this section, we compare different embeddings of physiological signals as discussed in Table 2 (where $Min^h$ represents PHASE). Based on these comparisons, we see the overall performance of predicting three adverse clinical events in Section 4.2.1 as well as a discussion of how well the embedding learners transfer between hospitals in Section 4.2.2. Lastly, in Section 4.2.3, we depict the attributions from our new model stacking method for obtaining Shapley values. In the operating room, there are a few physiological signals that stand out as indicators of adverse outcomes. Three of these signals include $SaO_2$ (blood oxygen), $ETCO_2$ (end tidal CO2), and NIBPM (non-invasive blood pressure measurement), which are linked to our three adverse outcomes: hypoxemia, hypocapnia, and hypotension, respectively. Forecasting these outcomes is particularly important because deviations from the norm could spell disaster (Lundberg et al., 2017; Curley et al., 2010; Barak et al., 2015). More details on labels in Section 6.1.

Table 2: Notation for different embeddings. Special notation includes: $Min^{h_1 \rightarrow h_2}$ and $Hypox^{h_1 \rightarrow h_2}$ means that the best model trained on hospital $h_1$ data is trained on $h_2$ data until convergence.

| Raw | Sixty minutes of raw signal. |
|---|---|
| EMA (Prescience) | Exponential moving averages as well as variances of each signal. Computed where weights decay with a half-life of 6 seconds, 1 minute, or 5 minutes. |
| $Min^h$ (PHASE) | Hidden layer embedding from an LSTM trained to predict the minimum of the current signal five minutes into the future on hospital h's data. |
| $Auto^h$ | Hidden layer embedding from an LSTM trained to predict an output signal identical to the input signal on hospital h's data. |
| $Hypox^h$ | Hidden layer embedding from an LSTM trained to predict hypoxemia on hospital h's data. |

### 4.2.1 PREDICTION PERFORMANCE

In this section, we aim to investigate the performance gains PHASE embeddings offer over other embeddings. Figure 2 shows the performance of XGB with different representations of the same signals across three prediction tasks. In terms of pre-training for this experiment, there is none for the Raw and EMA embeddings. However for $Min^h$ and $Auto^h$, we have trained fifteen univariate LSTM networks for both objectives across both hospitals (a total of sixty networks, not including fine-tuned networks). We fix these same LSTM networks to generate hidden embeddings of the original signals across the three final prediction tasks of hypoxemia, hypocapnia, and hypotension.

In terms of performance, the average precision points with all signals (in blue) are almost always significantly above their associated average precision points using only a single signal (in red). This suggests the outcomes derived from forecasting a single signal are complex and benefit from having access to more signals. Most importantly, with the exception of the fine tuned model (14), the $Min^h$ (PHASE) models (10 and 12) consistently outperform all other models in Figure 2 by a significant margin. The fact that LSTM autoencoders fail to grasp meaningful representations of the data in comparison to partially supervised LSTMs offers an insight; on our data set, the key to performant embeddings is to have closeness between the LSTM's prediction task and the downstream prediction task. In this case, effective embeddings for forecasting adverse outcomes related to too-low signals, required that the embeddings themselves were related to low signals in the future.

### 4.2.2 TRANSFERENCE BETWEEN HOSPITALS

In this section, we have two aims: the first aim is to evaluate how well our embedding models transfer. The second aim is to explore methods to repurpose them when there is a large amount of domain shift between hospitals. First, we can look back to Figure 2. The feature embeddings learned in a source hospital that differs to the target hospital (12) performs significantly better than the EMA (Prescience - Lundberg et al. (2017)) and Raw embeddings (2 and 4) and generally on par with a matching source and target hospital (10). This is promising, because it suggests that the domain shift between hospitals 0 and 1 does not prevent physiological signal embeddings from transferring well. Although these hospitals do have similar patient distributions, this transference is better than

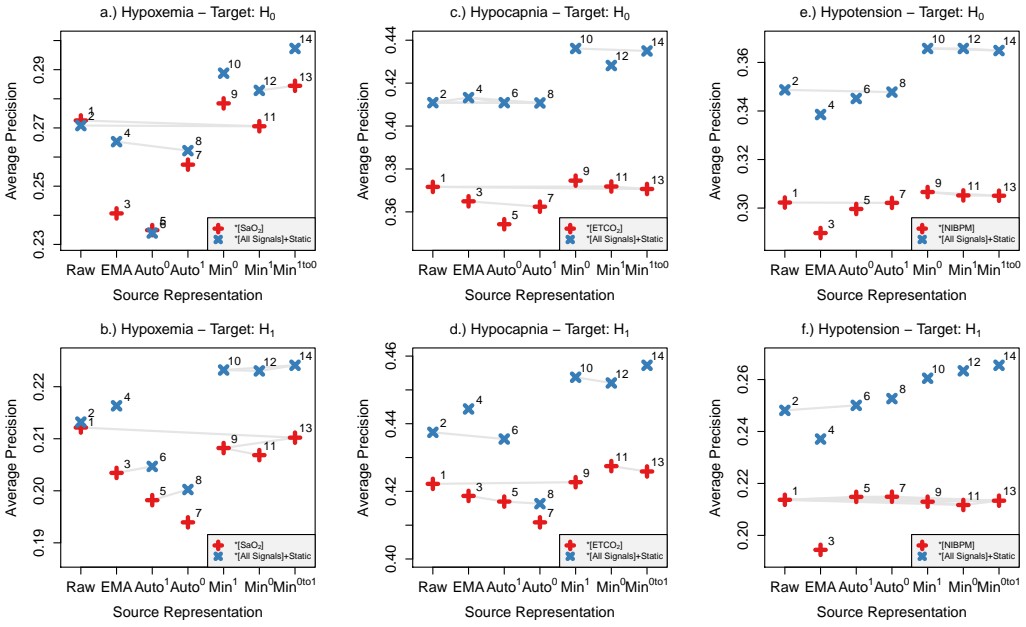

Figure 2: GBMs with different embeddings of physiological signals. Gray lines signify insignificant differences (all others pairs are significant at a p-value of 0.01) based on one hundred bootstraps of the test set with adjusted pairwise comparisons via ANOVA with Tukey's HSD test. For all, we utilize the 15 features above the line in both hospitals (Figure 5). Notation described in Table 2, where $Min^h$ represents PHASE. Note that $*[SaO_2]$ denotes that we have a Raw, EMA, or Min embedding of $SaO_2$ and $*[All Signals]+Static$ denotes a Raw, EMA, or Min embedding of all the signals plus static variables. More details about the setup for this experiment in Section 6.3, with p-values of models 10, 12, and 14 reported in Tables 7 and 8.

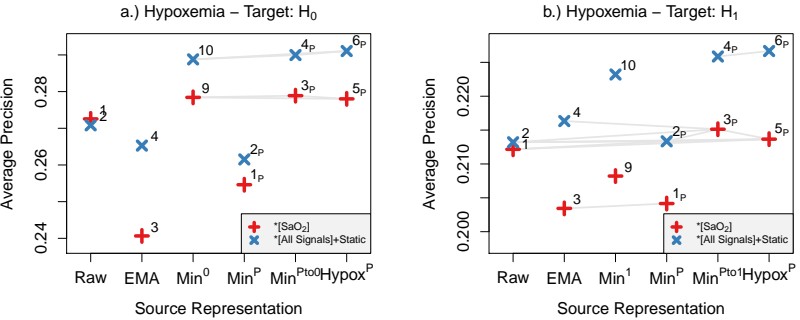

Figure 3: GBMs with different embeddings of physiological signals. Gray lines signify insignificant differences (all others are significant at a p-value of 0.01) based on one hundred bootstraps of the test set with adjusted pairwise comparisons via ANOVA with Tukey's HSD test. Notation described in Table 2. The PhysioNet embeddings borrow signals from the target hospital's embeddings so $*[SaO_2] + Min^T[Non SaO_2] + Static$ denotes that we have a $Min^P$ embedding of $SaO_2$, a $Min^T$ embedding of the remaining 14 variables, where T is the target hospital, and static variables. More details about the setup for this experiment in Section 6.4, with p-values reported in Tables 9 and 10.

might be expected, given that one hospital is an academic medical center and the other is a trauma center. As further evidence of their differences, we report the top ten diagnoses from each hospital in Appendix: Section 6.1 and find no overlap apart from CALCULUS OF KIDNEY between hospitals 0 and 1.

Next, in Figure 3, we can see that the $\text{Min}^P$ embeddings created from the publicly available PhysioNet data ($2_p$) are worse representations of $SaO_2$ in comparison to the embeddings trained with the target hospital's data (6). This result implies that the domain shift between hospital P and the OR hospitals is too large – resulting in learned Min embeddings ($2_P$) that are not as useful for prediction. Next, the improvement of the fine tuned models ($4_P$) over $\text{Min}^P$ embeddings, suggests the following insight: that fine tuning serves to recover the performance lost due to transference across distributionally disparate hospitals. Then, observing the improvement of $\text{Hypox}^P$ embeddings ($6_P$) over $\text{Min}^P$ embeddings, conveys a natural insight: that closeness between LSTM prediction tasks and the downstream prediction task is beneficial in the face of transference. This naturally extends the observation from Section 4.2.1: that closeness between LSTM prediction tasks and the downstream prediction task is beneficial in the face of performance. These two observations suggest two approaches to transferring across substantially different hospital data sets: 1.) train LSTMs with very specific prediction tasks that match the downstream prediction and 2.) fine tune LSTM networks.

### 4.2.3 Quantitative Validation of Interpretation for Stacked Models

In this section, our aim is to evaluate the efficacy of our interpretability method for stacked models. To do so, we separate interpretability evaluation approaches into two categories: qualitative and quantitative. Qualitative evaluations are important to ensure human insight into interpretability methods. However, our primary goal is to ensure that our novel method to obtain local feature attributions for stacked models is correct in the sense that the attributions estimate Shapley values. One qualitative evaluation of feature attributions is Lundberg et al. (2017) who demonstrate that local feature attributions improve the performance of practicing anesthesiologists in forecasting hypoxemia.

Our quantitative validation is a standard ablation/perturbation test, in a similar fashion to other interpretability evaluations: Arras et al. (2017), Hooker et al. (2018), Ancona et al. (2018), and Samek et al. (2017). The test consists of the following. For a single sample, we sort the input features according to their attributions, and iteratively impute each feature by the mean of the last two minutes. In order to ensure our interpretations generalize, we evaluate on the test set. Additionally, we use the top 1000 positive samples sorted by the predicted probability of hypoxemia (true positives). Then we evaluate the mean predicted probability across all samples, which will start high (for true positives) and monotonically decrease as we impute features, leading to an overall decrease in the average probability. Good interpretability methods should result in an *initial steepness* because the most "important" hypoxemic pathology is imputed first.

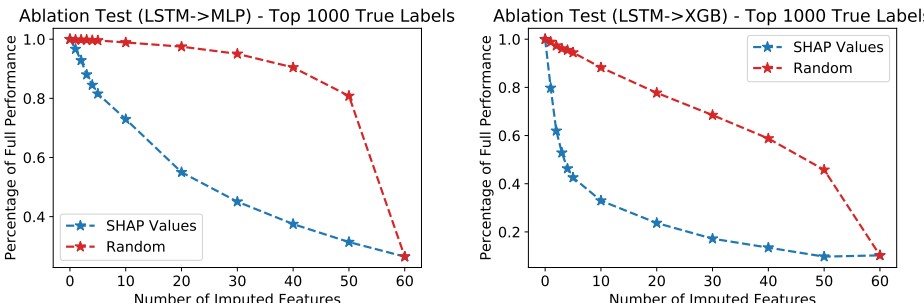

Figure 4: Ablation test on the top 1000 positive labels, sorted by the probability prediction of the final model. We "remove" the features (by imputing the mean of the last two minutes) according to Shapley values or a random ordering and then predict the probability of hypoxemia on the entirety of our test set. We obtain Shapley values for both models with a fixed background set of 100 samples. The model stack used for the interpretability evaluation (right) is (9) in Figure 2a. More details about the setup for this experiment in Section 6.5

Deep SHAP was originally proposed for traditional deep learning models, we extend it to support a mixture of model types in PHASE. As such, our primary aim is to evaluate against the pre-existing Deep SHAP methodology. We compare LSTM embeddings fed into XGB (PHASE) against LSTM embeddings fed into an MLP, because original Deep SHAP supports only traditional neural network architecture (see Figure 11 in the Appendix for more details on the model setups). In Figure 4, we

verify that ordering feature imputation by our interpretability method which combines Deep SHAP with Independent Tree SHAP (LSTM→XGB) does lead to initial steepness of the predicted probability of hypoxemia in a similar fashion to Deep SHAP alone (LSTM→MLP), with both methods outperforming a random ordering. In fact, it appears that for the Figure 4 (right), imputing our attributions for LSTM→XGB has a potentially more destructive effect on performance in the early number of imputed features. Lastly, in Appendix Section 6.5.1, we show attributions from both methods in a informal visual evaluation to confirm concordance.

## 5 CONCLUSION

This paper presents PHASE, a new approach to machine learning with physiological signals based on transferring embedding learners. PHASE has potentially far-reaching impacts, because neural networks inherently create an embedding before the final output layer. As discussed in Section 2.2, there is a large body of research independently working on neural networks for physiological signals. PHASE offers a potential method of collaboration by analyzing partially supervised univariate networks as semi-private ways to share meaningful signals without sharing data sets.

In the results section we offer several insights into transference of univariate LSTM embedding functions. First, closeness of upstream (LSTM) and downstream prediction tasks is indeed important for both predictive performance and transference. For performance, we found that predicting the minimum of the future five minutes was sufficient for the LSTMs to generate good embeddings. For transference, predicting the minimum of the next five minutes was sufficient to transfer across similar domains (operating room data from an academic medical center and a trauma center) when predicting hypoxemia. However when attempting to utilize a representation from Hospital P, we found that the difference between operating rooms and intensive care units was likely too large to provide good predictions. Two solutions to this include fine tuning the Min LSTM models as well as acknowledging the large amount of domain shift and training specific LSTM embedding models with a particular downstream prediction in mind. Last but not least, this paper introduced a way to obtain feature attributions for stacked models of neural networks and trees. By showing that Shapley values may be computed as the mean over single reference Shapley values, this model stacking framework generalizes to all models for which single reference Shapley values can be obtained, which was quantitatively verified in Section 4.2.3.

We intend to release code pertinent to training the LSTM models, obtaining embeddings, predicting with XGB models, and model stacking feature attributions – submitted as a pull request to the SHAP github (https://github.com/slundberg/shap). Additionally, we intend to release our embedding models, which we primarily recommend for use in forecasting "hypo" predictions.

In the direction of future work, it is important to carefully consider representation learning in health – particularly in light of model inversion attacks as discussed in Fredrikson et al. (2015). To this end, future work in making precise statements about the privacy of models deserves attention, for which one potential avenue may be differential privacy (Dwork, 2008). Other important areas to explore include extending these results to higher sampling frequencies. Our data was sampled once per minute, but higher resolution data may beget different neural network architectures. Lastly, further work may include quantifying the relationship between domain shifts in hospitals and PHASE and determining other relevant prediction tasks for which embeddings can be applied (e.g., "hyper" predictions, doctor action prediction, etc.).

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

# 6 APPENDIX

## 6.1 EXPERIMENTAL SETUP

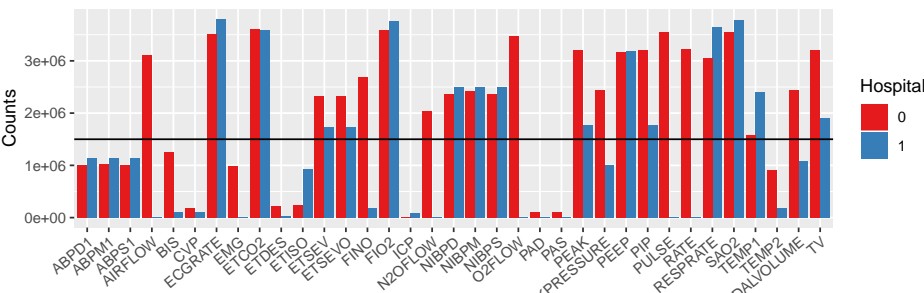

Figure 5: Counts of each feature across both AIMS hospitals. Fifteen features have more than 1.5 million counts for both hospitals (ECGRATE, ETCO2, ETSEV, ETSEVO, FIO2, NIBPD, NIBPM, NIBPS, PEAK, PEEP, PIP, RESPRATE, SAO2, TEMP1, TV).

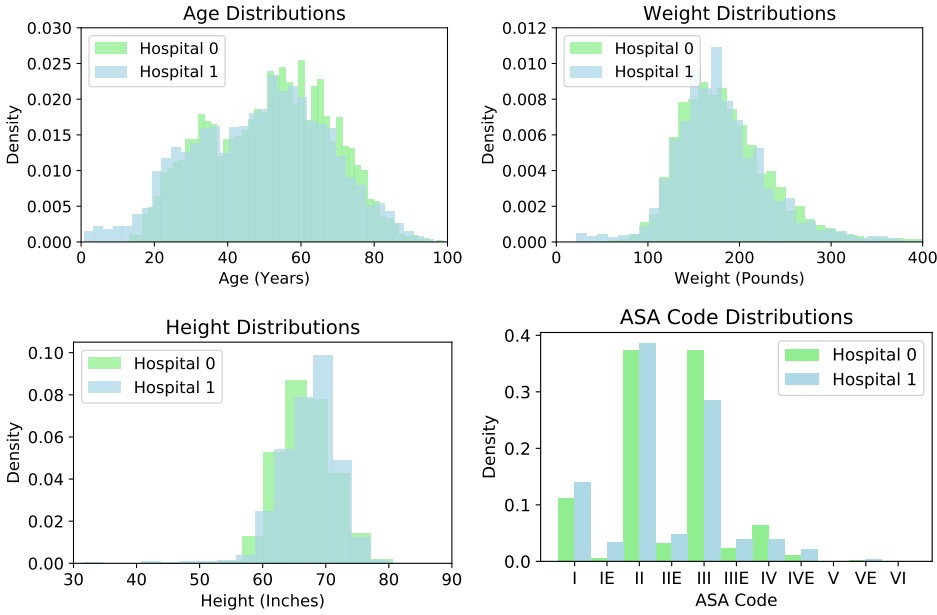

Figure 6: Differences between hospitals 0/1's distributions. One of the biggest difference is in sex. Hospital 0's data had $\approx 58\%$ female patients and hospital 1's data had $\approx 39\%$ female patients. Also, hospital 1 serves more young patients and only hospital 1 deals with ASA codes of VI.

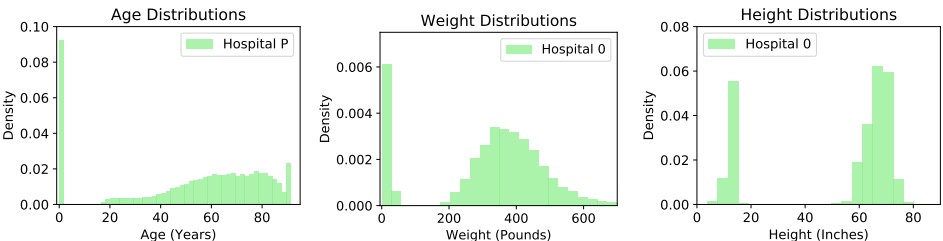

Figure 7: Distributions for hospital P. $\approx 44\%$ of patients are female. The skewed distributions are due to many newborn patients included in the dataset.

Top ten diagnoses (Hospital 0):

```
CATARACT NOS, SUBARACHNOID HEMORRHAGE, CALCULUS OF KIDNEY, OTHER
COMPLICATIONS DUE TO OTHER INTERNAL ORTHOPEDIC DEVICE IMPLANT
AND GRAFT, SENILE CATARACT NOS, SENILE CATARACT UNSPECIFIED,
NECROTIZING FASCIITIS, CATARACT, CARPAL TUNNEL SYNDROME, CMP NEC
D/T ORTH DEV NEC
```

Top ten diagnoses (Hospital 1):

```
MALIGNANT NEOPLASM OF BREAST (FEMALE) UNSPECIFIED, MALIGN NEOPL
BREAST NOS, ATRIAL FIBRILLATION, MORBID OBESITY, CALCULUS OF
KIDNEY, ESOPHAGEAL REFLUX, MALIGNANT NEOPLASM OF PROSTATE,
MALIGNANT NEOPLASM OF BLADDER PART UNSPECIFIED, PREV C-SECT
NOS-DELIVER, END STAGE RENAL DISEASE
```

Top ten diagnoses (Hospital P):

```
NEWBORN, PNEUMONIA, TELEMETRY, SEPSIS, CONGESTIVE HEART FAILURE,
CORONARY ARTERY DISEASE, CHEST PAIN, GASTROINTESTINAL BLEED,
ALTERED MENTAL STATUS, INTRACRANIAL HEMORRHAGE
```

**Labels** For hypoxemia, a particular time point $t$ is labelled to be one if the minimum of the next five minutes is hypoxemic $(\min(\text{SaO}_2^{t+1:t+6}) \leq 92)$. All points where the current time step is currently hypoxemic are ignored $(\text{SaO}_2^t \leq 92)$. Additionally we ignore time points where the past ten minutes were all missing or the future five minutes were all missing. Hypocapnia and hypotension are only labelled for hospitals 0 and 1. Additionally, we have stricter label conditions. We labeled the current time point $t$ to be one if $(\min(\mathcal{S}^{t-10:t}) > \mathcal{T})$ and the minimum of the next five minutes is "hypo" $(\min(\mathcal{S}^{t+1:t+5}) \leq \mathcal{T})$. We labeled the current time point $t$ to be zero if $(\min(\mathcal{S}^{t-10:t}) > \mathcal{T})$ and the minimum of the next ten minutes is not "hypo" $(\min(\mathcal{S}^{t+1:t+10}) > \mathcal{T})$. All other time points were not considered. For hypocapnia, the threshold $\mathcal{T} = 34$ and the signal $\mathcal{S}$ is ETCO$_2$. For hypotension the threshold is $\mathcal{T} = 59$ and the signal $\mathcal{S}$ is NIBPM. Additionally we ignore time points where the past ten minutes were all missing or the future five minutes were all missing. As a result, we have different sample sizes for different prediction tasks (reported in Table 4). For Min predictions, the label is the value of $\min(\mathcal{S}^{t+1:t+5})$, points without signal for in the future five minutes are ignored. For Auto predictions, the label is all the time points: $\mathcal{S}^{t-59:t}$. The sample sizes for Min and Auto are the same and are reported in Table 3.

Table 3: Sample sizes for the Min and Auto predictions for training the LSTM autoencoders. For the autoencoders we utilize the same data, without looking at the labels. We only utilize the 15 features above the line in both hospitals (Figure 5) for training our models.

| | Hospital 0 | | | Hospital 1 | | |
|---|---|---|---|---|---|---|
| Prediction Task | Train | Val | Test | Train | Val | Test |
| ECGRATE | 3.50e6 | 3.89e5 | 4.88e5 | 3.81e6 | 4.23e5 | 5.68e5 |
| ETCO2 | 3.60e6 | 4.00e5 | 5.01e5 | 3.59e6 | 3.99e5 | 5.37e5 |
| ETSEV | 2.32e6 | 2.57e5 | 3.27e5 | 1.73e6 | 1.93e5 | 2.61e5 |
| ETSEVO | 2.32e6 | 2.57e5 | 3.27e5 | 1.73e6 | 1.93e5 | 2.61e5 |
| FIO2 | 3.59e6 | 3.99e5 | 4.99e5 | 3.77e6 | 4.18e5 | 5.61e5 |
| NIBPD | 2.37e6 | 2.63e5 | 3.34e5 | 2.49e6 | 2.77e5 | 3.80e5 |
| NIBPM | 2.41e6 | 2.68e5 | 3.40e5 | 2.49e6 | 2.77e5 | 3.79e5 |
| NIBPS | 2.37e6 | 2.63e5 | 3.34e5 | 2.49e6 | 2.77e5 | 3.80e5 |
| PEAK | 3.20e6 | 3.56e5 | 4.47e5 | 1.77e6 | 1.96e5 | 2.63e5 |
| PEEP | 3.17e6 | 3.53e5 | 4.42e5 | 3.19e6 | 3.54e5 | 4.76e5 |
| PIP | 3.20e6 | 3.56e5 | 4.47e5 | 1.77e6 | 1.96e5 | 2.63e5 |
| RESPRATE | 3.06e6 | 3.40e5 | 4.27e5 | 3.65e6 | 4.05e5 | 5.44e5 |
| SAO2 | 3.55e6 | 3.94e5 | 4.96e5 | 3.79e6 | 4.21e5 | 5.65e5 |
| TEMP1 | 1.58e6 | 1.76e5 | 2.16e5 | 2.41e6 | 2.68e5 | 3.62e5 |
| TV | 3.21e6 | 3.57e5 | 4.47e5 | 1.90e6 | 2.11e5 | 2.84e5 |

Table 4: Sample sizes for the final downstream predictions.

| Prediction Task | Hospital 0 | | | Hospital 1 | | |
|---|---|---|---|---|---|---|
| | Train | Val | Test | Train | Val | Test |
| Hypoxemia | 3.53e6 | 3.92e5 | 4.93e5 | 3.75e6 | 4.17e5 | 5.60e5 |
| Hypocapnia | 1.13e6 | 1.26e5 | 1.58e5 | 1.58e6 | 1.75e5 | 2.32e5 |
| Hypotension | 1.65e6 | 1.83e5 | 2.35e5 | 2.10e6 | 2.33e5 | 3.20e5 |

Table 5: Base rates of different predictions in the test sets.

| Prediction Task | Hospital 0 | Hospital 1 |
|---|---|---|
| Hypoxemia | 1.06% | 2.32% |
| Hypocapnia | 9.91% | 7.94% |
| Hypotension | 7.41% | 3.52% |

Table 6: Sample sizes for hospital P. We don't use hospital P for a test set, so we only have training and validation sets. The base rate of hypoxemia on the validation set is 1.95%.

| Prediction Task | Train | Val |
|---|---|---|
| Hypoxemia/min5 | 4.57e6 | 5.08e5 |

## 6.2 Model Architecture and Training

**LSTM Architecture and Training:** We utilize LSTMs with forget gates, introduced by Gers et al. (2000), implemented in the Keras library with a Tensorflow back-end. We train our networks with either regression (Auto and Min embeddings) or classification (Hypox) objectives. For regression, we optimize using Adam with an MSE loss function. For classification we optimize using RMSProp with a binary cross-entropy loss function (additionally, we upsample to maintain balanced batches during training). Our model architectures consist of two hidden layers, each with 200 LSTM cells with dense connections between all layers. We found that important steps in training LSTM networks for our data are to impute missing values by the training mean, standardize data, and to randomize sample ordering prior to training (allowing us to sample data points in order without replacement). To prevent overfitting, we utilized dropouts between layers as well as recurrent dropouts for the LSTM nodes. Using a learning rate of 0.001 gave us the best final results. The LSTM models were run to convergence (until their validation accuracy did not improve for five rounds of batch stochastic gradient descent). In order to train these models, we utilize three GPUs (GeForce GTX 1080 Ti graphics cards).

**GBM Architecture and Training:** We train GBM trees in Python using XGBoost, an open source library for gradient boosting trees. XGBoost works well in practice in part due to it's ease of use and flexibility. Imputing and standardizing are unnecessary because GBM trees are based on splits in the training data, implying that scale does not matter and missing data is informative as is. We found that a learning rate of 0.02 for hypoxemia (0.1 for hypotension and hypocapnia), a max tree depth of 6, subsampling rate of 0.5, and a logistic objective gave us good performance. All XGB models were run until their validation accuracy was non-improving for five rounds of adding estimators (trees). In order to train these models, we utilize 72 CPUs (Intel(R) Xeon(R) CPU E5-2699 v3 @ 2.30GHz)

## 6.3 PREDICTION PERFORMANCE

Figure 8: *Model setup for Figure 2.* Showcasing most multivariate models in Figure 2, apart from the fine tuned LSTMs (14), which were trained identically but were initialized with the non-target hospital's corresponding LSTM. This figure is for hypoxemia, but hypocapnia and hypotension parallel this setup. Not counting fine tuned LSTMs, there are a total of 60 LSTMs: 30 Auto models for hospitals 0/1 and 30 Min models for hospitals 0/1. LSTM/XGB architecture and hyperparameters are consistent across models and can be found in Section 6.2. The signals and the outputs of the LSTMs are vectors. Multiple connections into a single model are simply concatenated. For all LSTMs, they consist of two layers each with 200 LSTM cells, trained in identical manners, as described in Section 6.2. For XGB, the training is detailed in Section 6.2 as well. The univariate predictions made in Figure 2 are similarly obtained, but only utilize the single feature used to obtain the final prediction. Here, "Hypoxemia" means: "Is $\min(\text{SaO2}_{(t+1,\cdots,t+5)}) \leq 92$?".

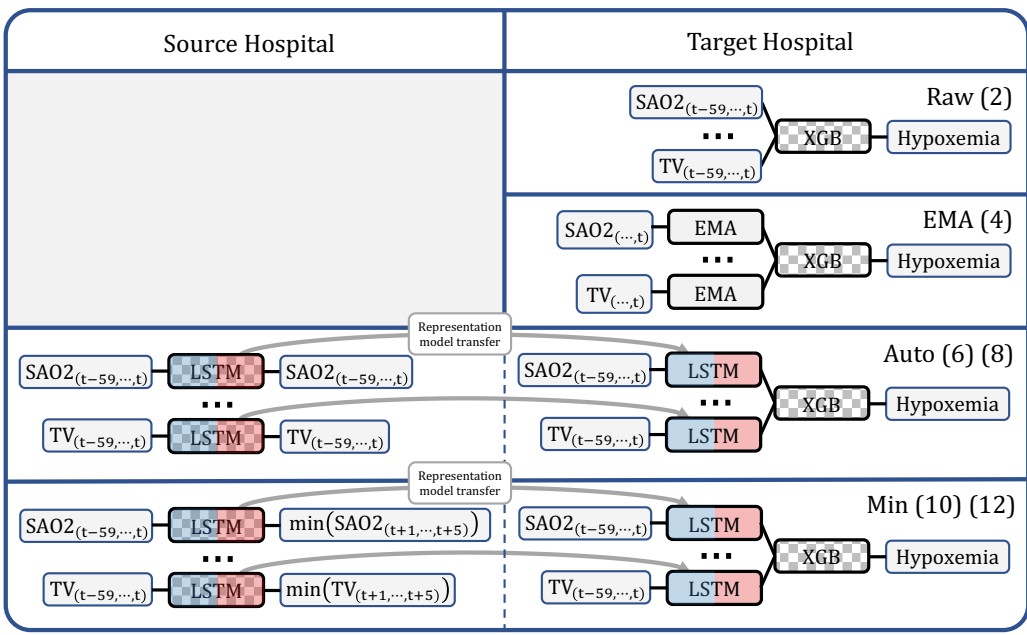

Table 7: *Hospital 0 p-values for Figure 2.* Reporting adjusted p-values based on one hundred bootstraps of the test set with adjusted pairwise comparisons via ANOVA with Tukey's HSD test. $0.0$ denotes a p-value less than $1e-14$. Since we are most concerned with models 10, 12, and 14 we only report pairs that include these models for the sake of brevity.

| | Fig 2a | | | Fig 2c | | | Fig 2e | | |
|---|---|---|---|---|---|---|---|---|---|
| | 10 | 12 | 14 | 10 | 12 | 14 | 10 | 12 | 14 |
| 1 | 0.0 | 0.0 | 0.0 | 0.0 | 0.0 | 0.0 | 0.0 | 0.0 | 0.0 |
| 3 | 0.0 | 0.0 | 0.0 | 0.0 | 0.0 | 0.0 | 0.0 | 0.0 | 0.0 |
| 5 | 0.0 | 0.0 | 0.0 | 0.0 | 0.0 | 0.0 | 0.0 | 0.0 | 0.0 |
| 7 | 0.0 | 0.0 | 0.0 | 0.0 | 0.0 | 0.0 | 0.0 | 0.0 | 0.0 |
| 9 | 0.0 | 3.32e−05 | 0.0 | 0.0 | 0.0 | 0.0 | 0.0 | 0.0 | 0.0 |
| 11 | 0.0 | 0.0 | 0.0 | 0.0 | 0.0 | 0.0 | 0.0 | 0.0 | 0.0 |
| 13 | 7.9e−5 | 0.88 | 0.0 | 0.0 | 0.0 | 0.0 | 0.0 | 0.0 | 0.0 |
| 2 | 0.0 | 0.0 | 0.0 | 0.0 | 0.0 | 0.0 | 0.0 | 0.0 | 0.0 |
| 4 | 0.0 | 0.0 | 0.0 | 0.0 | 0.0 | 0.0 | 0.0 | 0.0 | 0.0 |
| 6 | 0.0 | 0.0 | 0.0 | 0.0 | 0.0 | 0.0 | 0.0 | 0.0 | 0.0 |
| 8 | 0.0 | 0.0 | 0.0 | 0.0 | 0.0 | 0.0 | 0.0 | 0.0 | 0.0 |
| 10 | | 2.21e−9 | 0.0 | | 0.0 | 0.91 | | 1.0 | 0.90 |
| 12 | | | 0.0 | | | 0.0 | | | 0.90 |

Table 8: *Hospital 1 p-values for Figure 2.* Reporting adjusted p-values based on one hundred bootstraps of the test set with adjusted pairwise comparisons via ANOVA with Tukey's HSD test. $0.0$ denotes a p-value less than $1e-14$. Since we are most concerned with models 10, 12, and 14 we only report pairs that include these models for the sake of brevity.

| | Fig 2b | | | Fig 2d | | | Fig 2f | | |
|---|---|---|---|---|---|---|---|---|---|
| | 10 | 12 | 14 | 10 | 12 | 14 | 10 | 12 | 14 |
| 1 | 0.0 | 0.0 | 0.0 | 0.0 | 0.0 | 0.0 | 0.0 | 0.0 | 0.0 |
| 3 | 0.0 | 0.0 | 0.0 | 0.0 | 0.0 | 0.0 | 0.0 | 0.0 | 0.0 |
| 5 | 0.0 | 0.0 | 0.0 | 0.0 | 0.0 | 0.0 | 0.0 | 0.0 | 0.0 |
| 7 | 0.0 | 0.0 | 0.0 | 0.0 | 0.0 | 0.0 | 0.0 | 0.0 | 0.0 |
| 9 | 0.0 | 0.0 | 0.0 | 0.0 | 0.0 | 0.0 | 0.0 | 0.0 | 0.0 |
| 11 | 0.0 | 0.0 | 0.0 | 0.0 | 0.0 | 0.0 | 0.0 | 0.0 | 0.0 |
| 13 | 0.0 | 0.0 | 0.0 | 0.0 | 0.0 | 0.0 | 0.0 | 0.0 | 0.0 |
| 2 | 0.0 | 0.0 | 0.0 | 0.0 | 0.0 | 0.0 | 0.0 | 0.0 | 0.0 |
| 4 | 0.0 | 0.0 | 0.0 | 0.0 | 0.0 | 0.0 | 0.0 | 0.0 | 0.0 |
| 6 | 0.0 | 0.0 | 0.0 | 0.0 | 0.0 | 0.0 | 0.0 | 0.0 | 0.0 |
| 8 | 0.0 | 0.0 | 0.0 | 0.0 | 0.0 | 0.0 | 0.0 | 0.0 | 0.0 |
| 10 | | 0.99 | 0.94 | | 0.16 | 7.2e−8 | | 8.9e−6 | 0.0 |
| 12 | | | 0.80 | | | 0.0 | | | 5.0e−3 |

**Leave One Signal Out Test:** In Figure 9, we create a simulated setting – when predicting each event, we excluded the corresponding physiological signal from our features. For example, we assumed that $SaO_2$ is not recorded when predicting hypoxemia. Under this setting, we must rely on the remaining signals to predict hypoxemia. This setting is a more unsupervised evaluation in the sense that our outcome is not derived from a signal we create an embedding for. As our results show (Figure 9), PHASEs outperformance is consistent for hypocapnia and hypotension. For hypoxemia, all representations perform poorly because predicting hypoxemia heavily relies on SaO2, leaving little signal for the remaining features. This is likely due in part to the low base rates of hypoxemia: $1.06\%$ in hospital 0 and $2.32\%$ in hospital 1. Investigating further, we found that the log loss of the Min embeddings was lower than other embeddings on the validation set, but not on the test set. This overfitting further suggests that there was little signal to be captured, causing simpler embeddings like EMA to be favored.

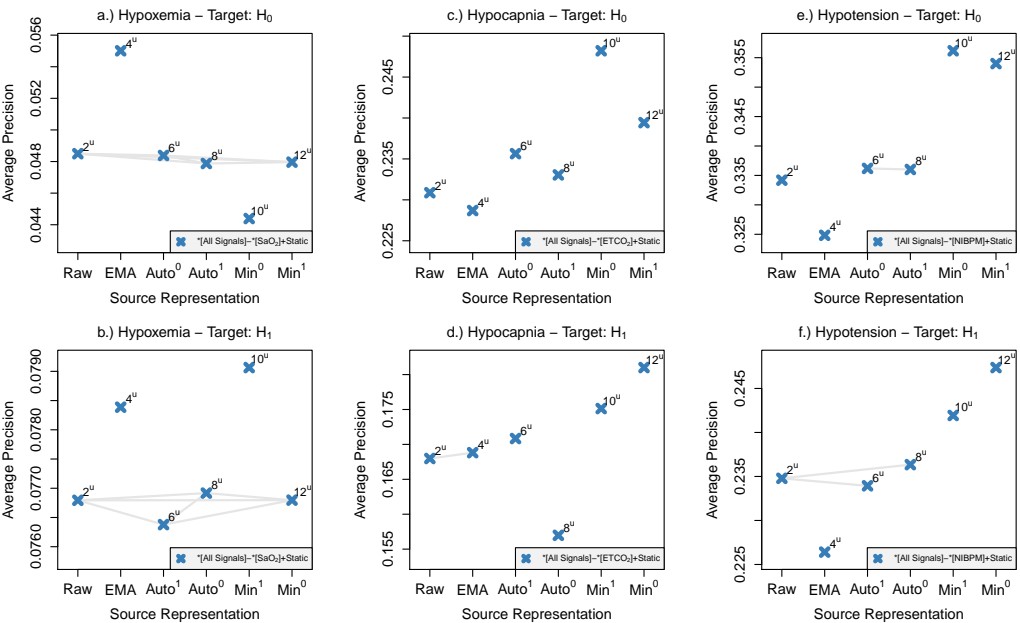

Figure 9: GBMs with different embeddings of physiological signals. Gray lines signify insignificant differences (all others pairs are significant at a p-value of 0.01) based on one hundred bootstraps of the test set with adjusted pairwise comparisons via ANOVA with Tukey's HSD test. For all, we utilize the 15 features above the line in both hospitals (Figure 5). Notation described in Table 2, where $Min^h$ represents PHASE. Note that $*[All Signals] - *[SaO_2]+Static$ denotes a Raw, EMA, or Min embedding of all the signals except for $SaO_2$ plus static variables.

## 6.4 TRANSFERENCE

Figure 10: *Model setup for Figure 3*. Showcasing what models are being used for the the transference experiment in Figure 3. LSTM/XGB architecture and hyperparameters are consistent across models and can be found in Section 6.2. The signals and the outputs of the LSTMs are vectors. Multiple connections into a single XGB model are simply concatenated. All LSTMs consist of two layers each with 200 LSTM cells, trained in identical manners, as described in Section 6.2. For XGB, the training is detailed in Section 6.2 as well. The univariate predictions made in Figure 3 are similarly obtained, but only utilize the single feature used to obtain the final prediction. Here, "Hypoxemia" means: "Is $\min(\text{SaO2}_{(t+1,\cdots,t+5)}) \leq 92$?".

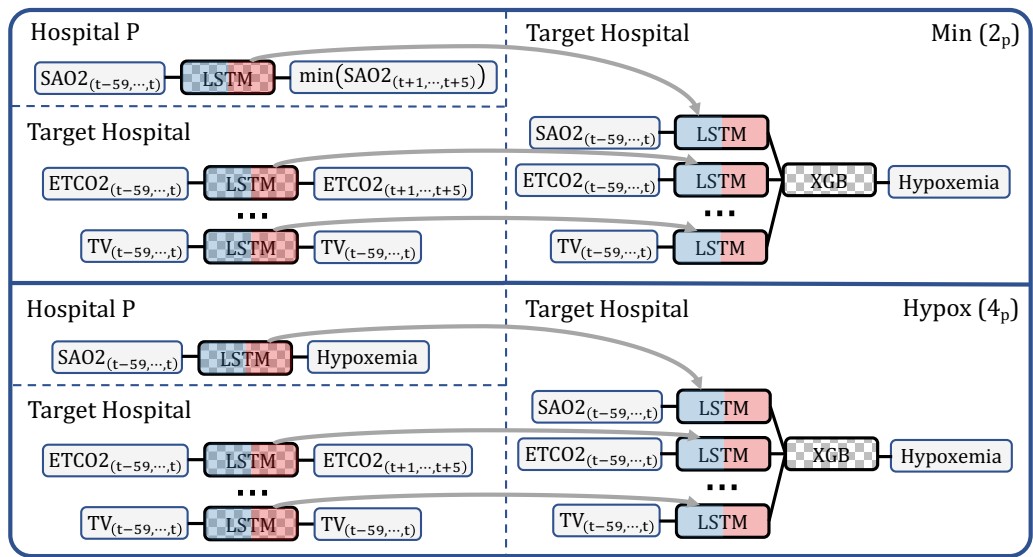

Table 9: *Hospital 0 p-values for Figure 3*. Reporting adjusted p-values based on one hundred bootstraps of the test set with adjusted pairwise comparisons via ANOVA with Tukey's HSD test.

|  | 1 | 3 | 5 | $1_P$ | $3_P$ | $5_P$ | 2 | 4 | 6 | $2_P$ | $4_P$ |
|---|---|---|---|---|---|---|---|---|---|---|---|
| 3 | 7e−13 | | | | | | | | | | |
| 5 | 9e−09 | 7e−13 | | | | | | | | | |
| $1_P$ | 7e−13 | 7e−13 | 7e−13 | | | | | | | | |
| $3_P$ | 3e−10 | 7e−13 | 1.000 | 7e−13 | | | | | | | |
| $5_P$ | 1e−07 | 7e−13 | 1.000 | 7e−13 | 0.998 | | | | | | |
| 2 | 0.736 | 7e−13 | 8e−13 | 7e−13 | 8e−13 | 1e−12 | | | | | |
| 4 | 1e−12 | 7e−13 | 7e−13 | 7e−13 | 7e−13 | 7e−13 | 8e−08 | | | | |
| 6 | 7e−13 | 7e−13 | 7e−13 | 7e−13 | 8e−13 | 7e−13 | 7e−13 | 7e−13 | | | |
| $2_P$ | 7e−13 | 7e−13 | 7e−13 | 4e−12 | 7e−13 | 7e−13 | 8e−13 | 0.002 | 7e−13 | | |
| $4_P$ | 7e−13 | 7e−13 | 7e−13 | 7e−13 | 7e−13 | 7e−13 | 7e−13 | 7e−13 | 0.979 | 7e−13 | |
| $6_P$ | 7e−13 | 7e−13 | 7e−13 | 7e−13 | 7e−13 | 7e−13 | 7e−13 | 7e−13 | 0.333 | 7e−13 | 0.988 |

Table 10: *Hospital 1 p-values for Figure 3.* Reporting adjusted p-values based on one hundred bootstraps of the test set with adjusted pairwise comparisons via ANOVA with Tukey's HSD test.

| | 1 | 3 | 5 | $1_P$ | $3_P$ | $5_P$ | 2 | 4 | 6 | $2_P$ | $4_P$ |
|---|---|---|---|---|---|---|---|---|---|---|---|
| 3 | 7e−13 | | | | | | | | | | |
| 5 | 1e−10 | 8e−13 | | | | | | | | | |
| $1_P$ | 7e−13 | 0.981 | 4e−11 | | | | | | | | |
| $3_P$ | 7e−06 | 7e−13 | 7e−13 | 7e−13 | | | | | | | |
| $5_P$ | 0.229 | 7e−13 | 8e−13 | 7e−13 | 0.260 | | | | | | |
| 2 | 0.780 | 7e−13 | 8e−13 | 7e−13 | 0.027 | 1.000 | | | | | |
| 4 | 9e−12 | 7e−13 | 7e−13 | 7e−13 | 0.574 | 1e−04 | 1e−06 | | | | |
| 6 | 7e−13 | 7e−13 | 7e−13 | 7e−13 | 7e−13 | 7e−13 | 7e−13 | 7e−13 | | | |
| $2_P$ | 0.603 | 7e−13 | 8e−13 | 7e−13 | 0.061 | 1.000 | 1.000 | 6e−06 | 7e−13 | | |
| $4_P$ | 7e−13 | 7e−13 | 7e−13 | 7e−13 | 7e−13 | 7e−13 | 7e−13 | 7e−13 | 1e−04 | 7e−13 | |
| $6_P$ | 7e−13 | 7e−13 | 7e−13 | 7e−13 | 7e−13 | 7e−13 | 7e−13 | 7e−13 | 4e−08 | 7e−13 | 0.953 |

## 6.5 INTERPRETABILITY

Figure 11: *Model setup for Figure 4.* Showcasing what models are being used to evaluate interpretability. LSTM/XGB architecture and hyperparameters are consistent across models and can be found in Section 6.2. The signals and the outputs of the LSTMs are vectors. Multiple connections into a single XGB model are simply concatenated. Here, "Hypoxemia" means: "Is $\min(\text{SaO2}_{(t+1,\cdots,t+5)}) \leq 92$?". Of special note, the MLP is trained identically to the Hypox models. The architecture is a single layer with 100 nodes with a relu activation connected densely into a sigmoid output node. The MLP is trained until convergence by upsampling the number of positive samples to match the negative samples for each batch. The attributions for LSTM→MLP are computed via Deep SHAP and the attributions for LSTM→XGB are computed via Deep SHAP combined with Independent Tree SHAP (our novel method). Both methods use a fixed background set of 100 randomly sampled points from the test set.

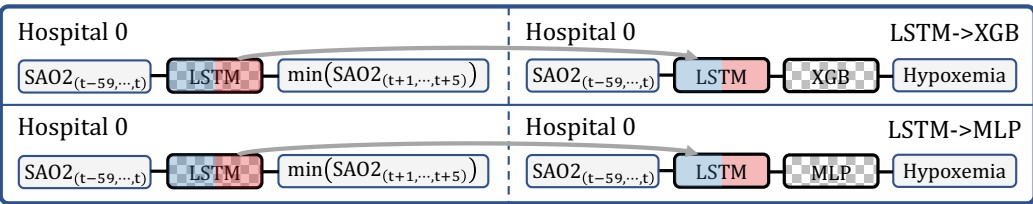

### 6.5.1 QUALITATIVE EVALUTION OF INTERPRETATION FOR STACKED MODELS

In Figure 12, we present the local feature attributions for two different model stacks across a random set of examples. We can see that the models generally appear to agree on their predictions, although there are occasional disagreements - which are likely due to the fact that these attributions are for two different models. Being able to observe these trends is useful to understanding models and to achieving credibility. As our primary aim is to ensure that our model stacking local feature attributions agree with feature attributions on neural networks, we also provide attributions for true positives and true negatives in settings where the models agree.

In Figure 13, we present local feature attributions for two different model stacks across true positive examples. Looking at these true positive examples we can see two consistent trends: high variability and a low absolute value of blood oxygen. Looking at the attributions we can discover that the dips in blood oxygen - minute to minute variability was important in both model stacks. Additionally, the closer the time point is to the actual prediction, the more important it is. In Figure 14, we present the local feature attributions for two different model stacks across true negative examples. Here we can see that variability and dips make a much smaller relative impact in the model predictions. Instead, the most important factor in determining hypoxemia is the high value of $\text{SaO}_2$ closest to the final prediction. Furthermore, the feature attributions reveal interesting trends in the attributions for the MLP model, where there appears to be a consistent trend even though the samples look fairly different.

Figure 12: *Randomly sampled feature attributions*. Local feature attribution plots for two stacked models: 1. LSTM→MLP and 2. LSTM→XGB. Here we present fifteen randomly sampled hypoxemia examples.

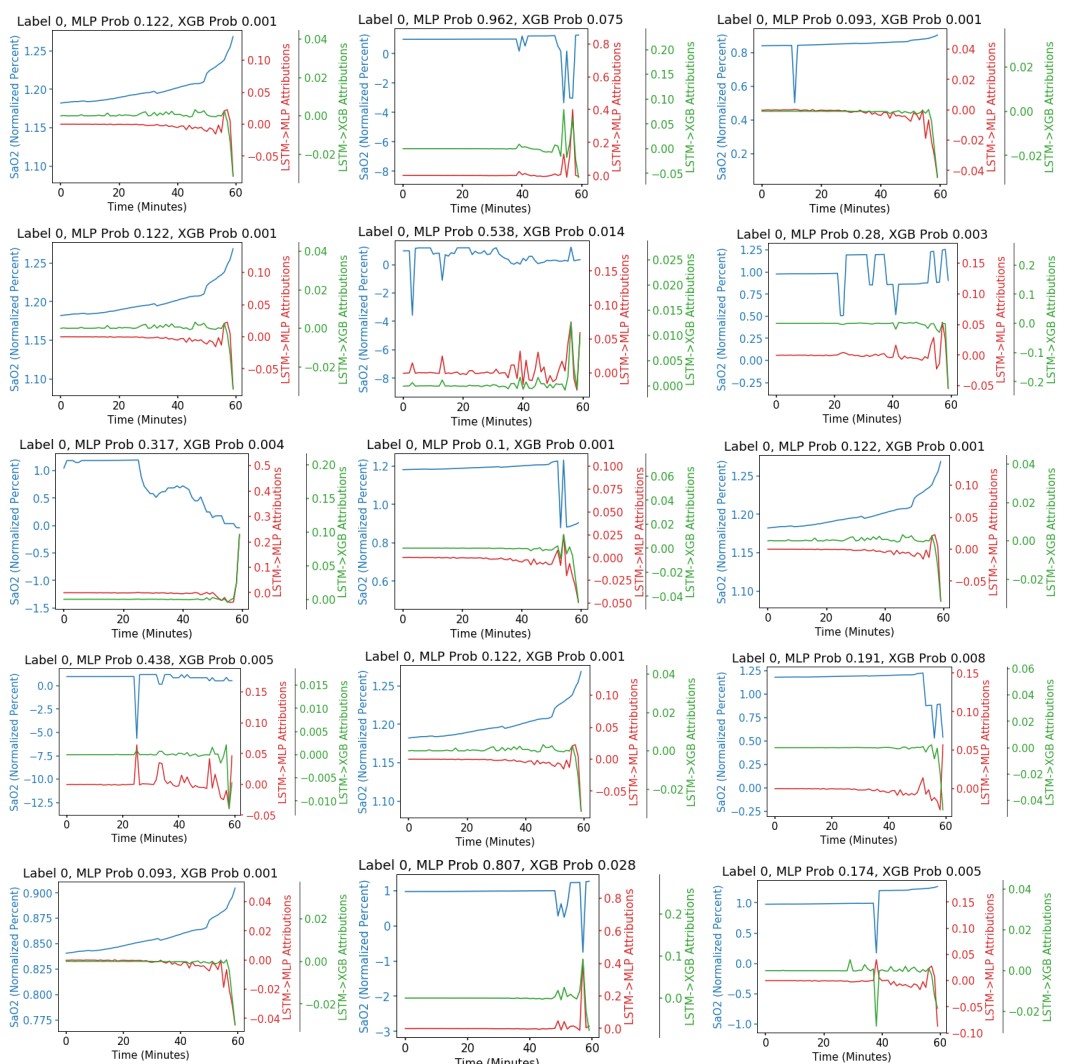

Figure 13: *True negative feature attributions*. Local feature attribution plots for two stacked models: 1. LSTM→MLP and 2. LSTM→XGB. Here we present the nine "least probable" positively labelled hypoxemia examples. In order to obtain this set, we took the intersection of the top 1000 negatively labelled examples from both models to get a set of 97 samples and randomly sample nine samples.

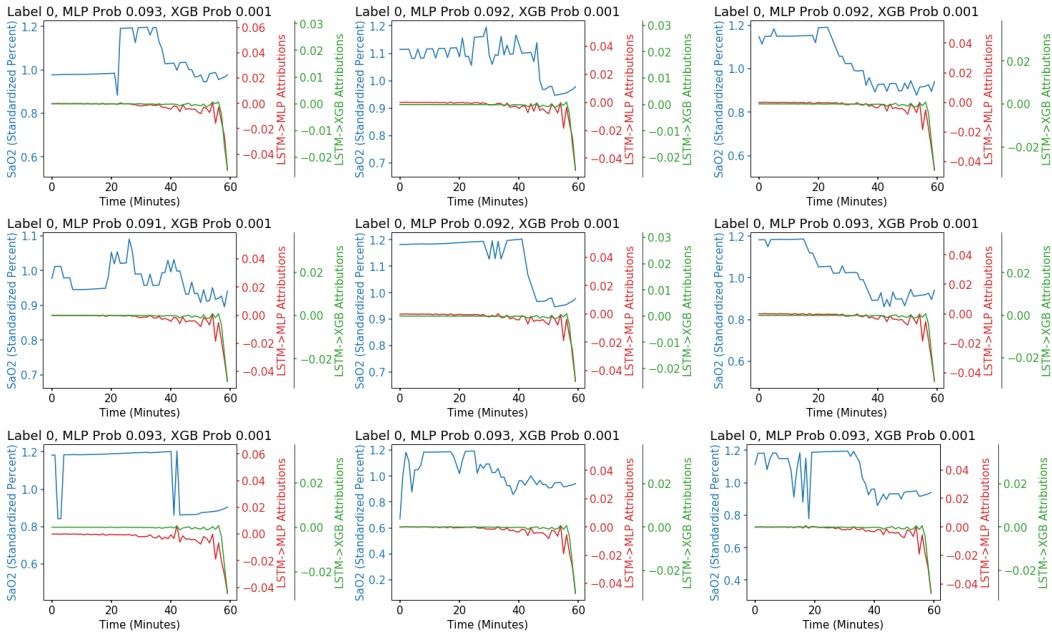

Figure 14: *True positive feature attributions*. Local feature attribution plots for two stacked models: 1. LSTM→MLP and 2. LSTM→XGB. Here we present the nine "most probable" positively labelled hypoxemia examples. In order to obtain this set, we took the intersection of the top 100 positively labelled examples from both models to get a set of 40 samples and randomly sample nine samples.

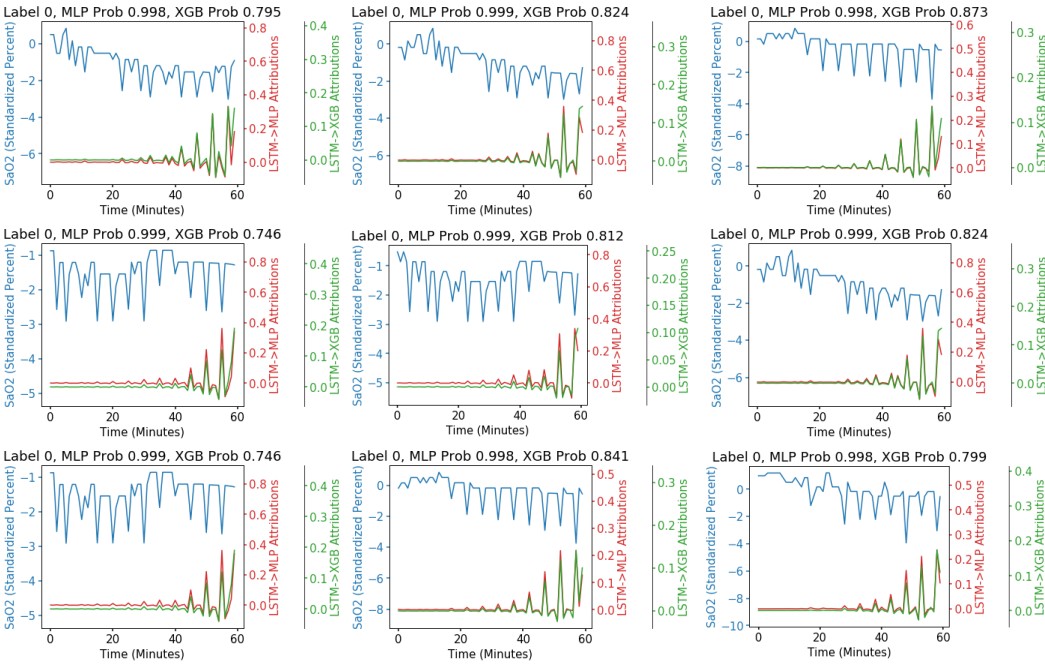

