# OpenReview forum: "Physiological Signal Embeddings (PHASE) via Interpretable Stacked Models"
_ICLR.cc/2019/Conference_

### Official Review · AnonReviewer2 · 2018-11-03
**This paper presents an approach to produce embeddings for physiological signals that are interpretable.**

**Rating:** 4
**Confidence:** 4

**Review:**

The authors claim contributions in three areas:
1) Learning representations on physiological signals. The proposed approach  uses LSTMS with a loss function that aims at predicting the next five minutes of the physiological signals. Based on their experiments, using this criteria outperforms
 LSTM autoencoder approaches that are tuned to reconstruct the original signals. The description of this work needs more details. It would be good to have clarity on these loss functions and also on the architecture of the LSTM autoencoder that is claimed here. Is it a standard seq2seq model? Is it something else?

2) They use the hidden state of the LSTMs as a representation of the inputs signals. From this representation, they have setup a set of supervised/predictive tasks to measure the efficacy of the representation. For this, they used gradient boosting machines.

3) They propose a way to estimate interpretability by tracking the impact of the input data on the predictions using an model agnostic approach using Shapley values. I have found this part of the paper particularly obscure. I recommend shedding some light on the structure of this model that generates these Shapley values.

The experimental result section also needs work in my opinion. First of all, the authors may want to better describe the data used. How many patients are in this set? How was the data partitioned for training, testing, validation? Any hyper-paremeter tuning? I have found the “transference” arguments a bit weak. First of all, the physical distance between hospital should not be mentioned as a way to compare “hospitals”. How did the authors select these features shown on Figure 2? MIMIC has more features than this. Why were these additional features discarded? Is the data coming from the same type of operating rooms in the case of hospital 0 and 1? I am somehow skeptical on the transfer of embeddings learned in an ICU setting to an OR setting. It would be great to provide details on the type of patients that are being monitored.

It is quite hard to argue from what’s presented in 4.3.3 that the proposed approach is interpretable. Can the authors explain how a visual inspection of Figure 5 “makes sense” as stated in the paper? What is the point that’s being made here? Any reason why more conventional attention mechanisms have not been looked at for interpretability?

Overall, I have found the problem addressed here interesting. However, I think that the paper needs work, both on the presentation of the methodology and also on the presentation of more convincing experimental arguments.

---

> ### Author Response · Authors · 2018-11-27
> **Rebuttal Part 1**
>
>
> We would like to thank the reviewers for their careful consideration of this manuscript and many suggestions for improvement. In response to the reviewers’ comments we have made changes that we feel substantially improve the manuscript and address the reviewers’ concerns, which we have responded to point-by-point.
>
> *“The description of this work needs more details… Is it something else? “
>
> We thank the reviewer for the beneficial feedback.  In terms of the loss functions. we utilize MSE loss for regression objectives (Min and Auto), whereas we use binary cross entropy for classification objects (Hypox).  The LSTM autoencoder is a seq2seq model with two layers with 200 LSTM cells each.  To better clarify our experimental setup, hyperparameter and model architecture details on the architectures of our LSTM and XGB models have been included in the Appendix: Section 6.2.
>
> *”3) They propose a way to estimate interpretability … I recommend shedding some light on the structure of this model that generates these Shapley values.”
>
> We thank the reviewer for highlighting this point of confusion.  Our objective is to validate our model stacked feature attributions on a straightforward univariate model (corresponding to model 9 in Figure 2a), which has been clarified in Section 4.2.3 along with a new quantitative evaluation of interpretability.  Appendix: Section 6.5 Figure 11 illustrates the model setup for the interpretability analysis.
>
> *”The experimental result section also needs work in my opinion … hyper-paremeter tuning?”
>
> We thank the reviewer for raising this important point.  Although the number of patients is reported in Table 1, the number of samples in each training, validation, and test set varies depending on the label being evaluated (which we have included in the Appendix: Section 6.1 Tables 3, 4, and 6).  Additionally we have added a paragraph describing the labelling methodology in more detail in Section 6.1 as well.  There was minimal hyperparameter tuning primarily due to the amount of models trained (90+ LSTMs and 108+ XGB models).  Instead, we have now included the final hyper parameter settings utilized in Appendix: Section 6.2.
>
> *“I have found the “transference” arguments a bit weak. First of all, the physical distance between hospital should not be mentioned as a way to compare “hospitals”. “
>
> We thank the reviewer for bringing up a great point.  Our aim was simply to suggest the distance between the hospitals might imply a domain shift without revealing the location of hospitals 0 and 1.  We have removed this point.  Instead, we describe in detail the distributions of statistics in each hospital - one being operating room data from a level 1 trauma center, one being operating room data from a university medical center, and the third one being waveform data from an ICU (in Appendix: Section 6.1, Figures 6 and 7).  Additionally, we report the top ten diagnoses from each hospitals in Appendix: Section 6.1.  We find no overlap apart from “CALCULUS OF KIDNEY” between hospitals 0 and 1.
>
> *“How did the authors select these features shown on Figure 2? MIMIC has more features than this. Why were these additional features discarded? “
>
> We thank the reviewer for the question.  To clarify Figure 5 (previously Figure 2), the features are from hospitals 0/1, not hospital P (MIMIC).  The hospital P dataset has a set of features that cover 7 (out of 15) features collected in hospitals 0/1. Therefore, incorporating MIMIC data to our experiments enables us to test PHASE’s ability to transfer relevant information from physiological signals in a challenging situation where hospitals have different features.
>
> Here, we chose to simplify our analysis and focus on hypoxemia, because (i) most of the signal for forecasting hypoxemia comes from SaO2 (which is also a feature that is consistently measured in hospital P), and (ii) we can test PHASE in a very challenging setting.  Additionally, this experimental setting has the added benefit of investigating whether the hospital P embeddings have interaction effects with hospital 0/1 embeddings.

---

> > ### Author Response · Authors · 2018-11-27
> > **Rebuttal Part 2**
> >
> >
> > *”Is the data coming from the same type of operating rooms ... provide details on the type of patients that are being monitored.”
> >
> > We thank the reviewer for addressing a point of confusion.  We have provided details on the type of patients being monitored for hospitals 0/1 as well as hospital P in Appendix: Section 6.1 Figures 6 and 7.  The distributions of hospitals 0/1 are far closer to each other than they are to hospital P.  Hospitals 0/1 differ primarily in the fact that one is a level 1 trauma center, whereas the other is a university medical center.
> >
> > Hospital P data is obtained from intensive care units, whereas hospital 0/1 data is obtained from operating rooms.  Another stark difference is that Hospital P data contains a great deal of data from newborns.  Although these datasets are quite different, using ICU data still makes sense for two reasons.  The first reason is simply that they may still capture something useful.  Similarly to computer vision, the learned LSTM embeddings may have lower-level and higher-level representations.  Although transferring the specific (higher-level) representations may not be useful, transferring some of the lower level ones may be.  The second reason is that even if the fixed LSTMs do not create performant embeddings, they may still contain useful information like model architectures and hyperparameters that are known to be good for predicting hypoxemia, for example.  Additionally, fine-tuning has been shown to outperform of match neural networks trained from scratch, with other benefits such as robustness to size of training sets (Tajbakhsh et. al. IEEE 2016).
> >
> > *“It is quite hard to argue from what’s presented in 4.3.3 that the proposed approach is interpretable. Can the authors explain how a visual inspection of Figure 5 “makes sense” as stated in the paper? What is the point that’s being made here?“
> >
> > We thank the reviewer for bringing up a great point.  Figure 5 (now removed) was primarily meant to serve as a sanity check that the proof for feature attributions we presented is correct.  Rather than explore anecdotal examples, our aim in the revision is to quantitatively ensure our local feature attributions are correct quantitatively on a straightforward univariate model (corresponding to model 9 in Figure 2a).  Our quantitative validation is a standard ablation/perturbation test, in a similar fashion to other interpretability evaluations: Arras et al. (arXiv 2017), Hooker et al. (arXiv 2018), Ancona et al. (ICLR 2018), and Samek et al. (IEEE 2017).   The test consists of the following.  For a single sample, we sort the input features according to their attributions, and iteratively impute each feature by the mean of the last two minutes.  In order to ensure our interpretations generalize, we evaluate on the test set.  Additionally, we use the top 1000 positive samples sorted by the predicted probability of hypoxemia (true positives).  Then we evaluate the mean predicted probability across all samples, which will start high (for true positives) and monotonically decrease as we impute features, leading to an overall decrease in the average probability.  Good interpretability methods should result in an initial steepness because the most "important" hypoxemic pathology is imputed first -- similar to Ancona et al. (2018).
> >
> > However, it is difficult to validate our model stacking feature attributions because no other attribution methods exist for stacks of neural networks and trees.  Instead, we replace the tree part of our model stack (LSTM->XGB) with a multi-layer perceptron, which creates a fully neural network model stack (LSTM->MLP) that we can directly apply an existing attribution method to.  Then, we would hope to see that our novel model stacking feature attribution method (DeepSHAP + Independent TreeSHAP on LSTM->XGB) provides attributions that performs similarly to the pre-existing method (DeepSHAP on LSTM->MLP) on the ablation test (shown in Figure 4).
> >
> > Lastly, we augment the previous quantitative assessment with a qualitative one.  We look at true positive, true negative, and randomly selected examples in Appendix: Section 6.5.1, accompanied by a brief discussion to visually demonstrate that our novel method for feature attributions matches with a more conventional approach applied to a fully neural network model.

---

> > > ### Author Response · Authors · 2018-11-27
> > > **Rebuttal Part 3**
> > >
> > >
> > > *“Any reason why more conventional attention mechanisms have not been looked at for interpretability?”
> > >
> > > We thank the reviewer for addressing a point of confusion.  There are two primary reasons why we did not use attention mechanisms for interpretability.
> > >
> > > 1.	The main reason is because our aim is to ensure interpretability for arbitrary downstream models. When sharing embeddings you don’t want to force the downstream user to use a specific model type. We enable a broad set of stacked models, where the model stack we propose combines a tree model (XGB) with neural network embeddings as features.  Because attention mechanisms are specific to particular kinds of neural networks, they cannot provide attributions for the entire stack.  We illustrate that Shapley values, which naively have exponential computational complexity, can be obtained in polynomial time for stacked models of arbitrary combinations of trees and neural network components.
> > > 2.	We chose Shapley values, because of their theoretical basis in game theory.  As shown in “A Unified Approach to Interpreting Model Predictions” (Lundberg et al. NIPS 2017), the Shapley values are the only solution that maintain three desirable properties for feature attributions: local accuracy, missingness, and consistency.
> > >
> > > *“Overall, I have found the problem addressed here interesting. However, I think that the paper needs work, both on the presentation of the methodology and also on the presentation of more convincing experimental arguments.”
> > >
> > > We thank the reviewer for helping us to improve our paper through better descriptions of our data, of the model architectures, of the feature learning process, and of the interpretability.  To support our argument that PHASE can provide interpretable explanations, we presented quantitative evaluation results based on a standard ablation test.

---

### Official Review · AnonReviewer3 · 2018-11-03
**Well-motivated, well-written, but some issues with the experiments.**

**Rating:** 5
**Confidence:** 4

**Review:**

Summary of the paper:
This paper proposes PHASE, a framework to learn the embeddings for physiological signals from medical records, which can be used in downstream prediction tasks, possibly across domains (i.e. different patient distribution). The authors employ separate LSTMs for each signal channel that are trained to predicts the minimum value of the signal in the fixed future time window (5 minutes in this paper). After training the LSTMs, the learned signal embeddings are fed to gradient boosted trees for a specific prediction task (e.g. predicting whether hypoxemia will occur in 5 minutes). Once the LSTMs are trained, they can be re-used for another dataset; the LSMTs are fixed, and generate embeddings that are fed to a new trainable gradient boosted trees for performing a similar task. The authors also combine existing attribution methods (DeepSHAP and Independent TreeSHAP) to provide some explanation of PHASE. The authors use three different datasets to test PHASE's prediction performance, transferability of the embeddings, and interpretation.

Pros:
- The paper is well-motivated, well-organized and clearly written. The reading experience was smooth.
- Given the importance of physiological signals in ICU settings, transferable embeddings can be an important technique in practice
- As the authors claim, I am not aware of any notable prior work on transferable physiological signal embeddings. The authors tackle a relatively unexplored territory.

Issues:
- The authors claim PHASE learns signal embeddings that are transferable. However, the authors train the embeddings to predict the minimum value within the next five time steps, because the downstream tasks are all predicting whether a certain signal goes below some threshold ("hypo"xenia, "hypo"capnia, "hypo"tension). This means the authors designed the embedding learning process with a priori knowledge of the downstream tasks, which significantly weakens theirs claim that PHASE learns transferable embeddings. Word embeddings trained on Wikipedia, or ConvNets trained on ImageNet are not designed to be used in a specific type of downstream tasks. What PHASE demonstrates is basically that "hypo"xxxx predictions can be accurately made with pre-training the embeddings to predict a very relevant task.
- The authors claim that transferred PHASE embeddings significantly outperform EMA or Raw. But I wouldn't call 0.005-0.02 AP improvement "significant". Model 12 in Figure 3 shows better performance than model 2 and 4, but the gap is not that large.
- More importantly, the fact that model 10 and model 12 show similar performance is not very surprising. The two hospitals are in the same city, only miles away. Naturally the distribution of the patients would not be too different. Given this, claiming that PHASE embeddings are transferable does not have a strong ground.
- The claim for transferable embedding is further weakened by Figure 4. Model 1^p in Figure 4 clearly performs worse than Raw, which means embeddings learned from significantly different setting (hospital P) is actually making it harder for XGB than simply looking at raw signals. If PHASE was learning a robust embeddings, then the learned embeddings should at least not hurt the performance of XGB.
- Evaluating the interpretation of the model is weak. All the authors did was pick four examples and provide qualitative explanation. And they do not even describe whether this interpretation is from model 9 or 10. It would have been much better if at least one medical expert took a look at more than a few examples. In the current form, we cannot be sure if the model is using the SaO2 signal in a medically meaningful way. Also, if this is the interpretation of model 10 or 12, then we should look at the attributions for other signals as well.
- Lack of description on experiment setup. The authors do not describe how they pre-trained the LSTMs to obtain Min^h, Auto^h and Hypox^h, which significantly hurts reproducibility. Also I couldn't find any description regarding train/test splits or cross validations, or size of the LSTM cells.
- More description is necessary as to how Raw was used to train XGB. Was the entire sequence of 15 signals fed to XGB?
- Y-axis of Figure 5 is not on the same scale. This makes it hard to intuitively understand the change of SaO2.

---

> ### Author Response · Authors · 2018-11-27
> **Rebuttal Part 1**
>
>
> We would like to thank the reviewers for their careful consideration of this manuscript and many suggestions for improvement. In response to the reviewers’ comments we have made changes that we feel substantially improve the manuscript and address the reviewers’ concerns, which we have responded to point-by-point.
>
> *”- The authors claim PHASE learns signal embeddings that are transferable. … to predict a very relevant task.”
>
> We thank the reviewer for bringing up an excellent point. When we considered prediction problems for our paper, we focused on largely two aspects: (i) clinical importance, and (ii) real-time prediction problems, which are an appropriate evaluation setting for time-series embedding methods. The outcomes we considered - hypoxemia, hypotension, and hypocapnia - are representative adverse real-time events caused by surgery complications and are a significant cause of anesthesia-related complications (Lundberg et al. Nature BME 2018, Barak et al. Sci. World. Journal, 2015; Curley et al. Crit. Care. Med., 2010).  As further justification, perioperative adverse outcomes are often due to signals that are too low in terms of magnitude (Exclamado et. al. The Laryngoscope 1989).  Therefore training models on the lower boundaries of signals (“hypo”) would, in all likelihood, cover a non-trivial group of important adverse outcomes.  Future work training “hyper” models as well as working with physicians to identify other such groupings of physiological prediction tasks would certainly be meaningful as well.
>
> Finally, to further address the reviewer’s comments, we have mitigated claims of PHASE being unsupervised and instead called our LSTM models “partially supervised” throughout the entirety of the manuscript.  We denote “partially supervised” to mean LSTMs trained with prediction tasks related to the final downstream prediction.  Furthermore, we have refined the discussion in Sections 4.2.1 Paragraph 2 and 4.2.2 Paragraph 3 to emphasize that completely unsupervised LSTMs (e.g., autoencoders) are insufficient for downstream “hypo” predictions, which are clinically important perioperative outcomes.  In fact, on our datasets, we found that closeness in the LSTM prediction tasks to the ultimate downstream prediction tasks is beneficial to performance as well as transference.  In order to change the message of our paper, we have added this to our conclusion as well (Section 5 Paragraph 2).  As a last note, we recommend our models for use in forecasting "hypo" predictions, a statement we have added to the conclusion (Section 5 end of Paragraph 3).

---

> > ### Author Response · Authors · 2018-11-27
> > **Rebuttal Part 2**
> >
> >
> > *”- The authors claim that transferred PHASE embeddings significantly outperform EMA or Raw. … the gap is not that large.”
> >
> > We thank the reviewer for bringing up a great point.  We argue that like other prediction problems in ML, the percentage improvement is often more relevant than absolute difference in AP. We believe that our results showing the improvement of PHASE over other representations are significant for the following reasons:
> >
> > 1.	Our intention was to show that PHASE embeddings improve over state-of-the-art prediction models even when using the same dataset. However, the largest clinical impact will likely come from sharing these embeddings, allowing embeddings from a large dataset to be used for a training problem in a smaller dataset. In these situations the AP gain of 0.04 is just a lower bound of the improvement we get from the method, much larger gains are possible when people use these to add power to smaller datasets.
> > 2.	In Lundberg et. al. (2018), the best performing model (analogous to model 4 in Figure 2) is able to achieve higher predictive accuracy than practicing anesthesiologists in predicting hypoxemia and increase doctors’ ability to forecast hypoxemia by providing Shapley value attributions.  With PHASE, we gain further improvement over Prescience (up to 11% improvement in AP - Figure 2f: Model 12 compared to Model 4) and provide Shapley value attributions for our stacked models, leaving little reason to prefer Prescience over PHASE in a hospital.  In health, this level of improvement in predictive performance may impact a number of patients over long periods of time, as pointed out in Lundberg et al. (2018).
> > 3.	The relative improvement of being able to increase precision across all recalls (by 5.6% on average over EMA and 4.6% on average over Raw) would mean substantially better retrieval of adverse outcomes, beneficial in the face of alarm fatigue and for patient care.  Additionally, the absolute improvement of 0.02 is fairly large given that AP ranges between 0 and 1.
> > 4.	Finally, Model 12 is shown to be statistically significantly better than competing models at a p-value of 0.01 based on one hundred bootstraps of the test set with adjusted pairwise comparisons via ANOVA with Tukey’s HSD test. Moreover, the p-values comparing Model 12 to all others were significant at a much lower threshold than 0.01 (often Model 12 is significantly better even at a threshold of 1e-10).  We have added these p-values for Figures 2 and 3 to the Appendix: Section 6.3 Tables 7 and 8 as well as Section 6.4 Tables 9 and 10.
> >
> > *”- More importantly, the fact that model 10 and model 12 show similar performance is not very surprising. … does not have a strong ground. “
> >
> > We thank the reviewer for bringing up a great point.  As Reviewer #2 said “the physical distance between hospital should not be mentioned as a way to compare hospitals,” we removed the discussion of the distance between hospitals 0/1.  Instead, one major difference between hospitals 0/1 is that one hospital is an academic medical center, whereas the other is a level 1 trauma center. This causes significant differences in the patient populations, which reflects in distributions that we illustrate in Appendix Section 6.1: Figure 6.  Additionally, we report the top ten diagnoses from each hospitals in Appendix: Section 6.1.  We find no overlap apart from “CALCULUS OF KIDNEY” between hospitals 0 and 1.  One such distributional shift is that Hospital 0 data had roughly 58% female patients and hospital 1 data had roughly 39% female patients.  Other differences include the fact that hospital 1 serves more young patients than hospital 0 and the fact that only hospital 1 deals with ASA codes of VI.
> >
> > For medical research, transferability results across two distinct hospitals have been considered very important (Wiens et. al. JAMIA 2014, Choi et. al. KDD 2016, Lee et. al. IEEE 2012).  Our results imply one potential model where a large medical center in a given state shares representation learners with small neighboring medical centers, boosting the smaller medical center’s capability to predict adverse outcomes without risking patient privacy.  Given that there is no prior work examining transference of embedding functions, even transference under a small domain shift is an important first step towards medical credibility.

---

> > > ### Author Response · Authors · 2018-11-27
> > > **Rebuttal Part 3**
> > >
> > >
> > > *”- The claim for transferable embedding is further weakened by Figure 4… hurt the performance of XGB.”
> > >
> > > We thank the reviewer for bringing up an excellent point.  In the revised paper, we show that Reviewer #1’s suggestion - fine tuning LSTM model (i.e., Model 4p) transferred from the source hospital - can remove this concern.  Figure 3 shows that even in this very challenging setting (i.e., extracting information from SaO2 signals in ICU), PHASE can extract relevant information from SaO2 signals.  We have applied this clarification to Section 4.2.2.
> > >
> > > *”- Evaluating the interpretation of the model is weak… as well.”
> > >
> > > We thank the reviewer for bringing up a great point.  Figure 5 (now removed) was primarily meant to serve as a sanity check that the proof for feature attributions we presented is correct.  Rather than explore anecdotal examples, our aim in the revision is to quantitatively ensure our local feature attributions are correct on a straightforward univariate model (corresponding to model 9 in Figure 2a).  Our quantitative validation is a standard ablation/perturbation test, in a similar fashion to other interpretability evaluations: Arras et al. (arXiv 2017), Hooker et al. (arXiv 2018), Ancona et al. (ICLR 2018), and Samek et al. (IEEE 2017).   The test consists of the following.  For a single sample, we sort the input features according to their attributions, and iteratively impute each feature by the mean of the last two minutes.  In order to ensure our interpretations generalize, we evaluate on the test set.  Additionally, we use the top 1000 positive samples sorted by the predicted probability of hypoxemia (true positives).  Then we evaluate the mean predicted probability across all samples, which will start high (for true positives) and monotonically decrease as we impute features, leading to an overall decrease in the average probability.  Good interpretability methods should result in an initial steepness because the most "important" hypoxemic pathology is imputed first similar to Ancona et al. (2018).
> > >
> > > However, it is difficult to validate our model stacking feature attributions because no other attribution methods exist for stacks of neural networks and trees.  Instead, we replace the tree part of our model stack (LSTM->XGB) with a multi-layer perceptron, which creates a fully neural network model stack (LSTM->MLP) that we can directly apply an existing attribution method to.  Then, we would hope to see that our novel model stacking feature attribution method (DeepSHAP + Independent TreeSHAP on LSTM->XGB) provides attributions that performs similarly or better to the pre-existing method (DeepSHAP on LSTM->MLP) on the ablation test (shown in Figure 4).
> > >
> > > Lastly, we augment the previous quantitative assessment with a qualitative one.  We look at true positive, true negative, and randomly selected examples in Appendix: Section 6.5.1, accompanied by a brief discussion to visually demonstrate that our novel method for feature attributions matches with a more conventional approach applied to a fully neural network model.

---

> > > > ### Author Response · Authors · 2018-11-27
> > > > **Rebuttal Part 4**
> > > >
> > > > *”- Lack of description on experiment setup. The authors do not describe how they pre-trained the LSTMs to obtain Min^h, Auto^h and Hypox^h, which significantly hurts reproducibility. Also I couldn't find any description regarding train/test splits or cross validations, or size of the LSTM cells.”
> > > >
> > > > We thank the reviewer for raising this important point.  The number of samples in each training, validation, and test set varies depending on the label being evaluated - we have included these samples sizes in the Appendix: Section 6.1 Tables 3, 4, and 6.  Additionally we have added a paragraph describing the methods for obtaining the labels in more detail in Section 6.1 as well.  To better clarify our experimental setup, more details on the architectures of our LSTM and XGB models have been added in the Appendix: Section 6.2.  As a final note, for reproducibility, we plan to release code pertinent to training the LSTM models, obtaining embeddings, predicting with XGB models, and model stacking feature attributions - submitted as a pull request to the SHAP github (https://github.com/slundberg/shap).  We have indicated our intent to do so in the conclusion (Section 5 Paragraph 3).  Additionally, we intend to release our embedding models, which we recommend for use in forecasting "hypo" predictions.
> > > >
> > > > *”- More description is necessary as to how Raw was used to train XGB. Was the entire sequence of 15 signals fed to XGB? “
> > > >
> > > > We fed in 60 minutes of the 15 Raw signals concatenated together.  To better clarify our experimental setup, more details on the architectures of our LSTM and XGB models have been added in the Appendix: Section 6.2.  Additionally, we have a more in-depth description of the experimental setup for each experimental result in Appendix: Section 6.3 Figure 8, Section 6.4 Figure 10, and Section 6.5 Figure 11.
> > > >
> > > > *”- Y-axis of Figure 5 is not on the same scale. This makes it hard to intuitively understand the change of SaO2.”
> > > >
> > > > We thank the reviewer for this feedback; however, because our primary aim is to evaluate Shapley values, of more importance to see changes in the signal rather than the absolute value.  As such, we scale for each example, rather than across all examples.  Additionally, we have moved Figure 5 to the Appendix: Section 6.5.1 and added many more examples and a brief discussion into the feature attributions provided for true positives, true negatives, and random samples with a fully neural network model (LSTM->MLP) and a hybrid one (LSTM->XGB).

---

### Official Review · AnonReviewer1 · 2018-11-03
**Transfer learning for physiological signals in the OR and ICU**

**Rating:** 6
**Confidence:** 5

**Review:**

The authors present a new method for learning unsupervised embeddings of physiological signals (e.g. time series data) in a healthcare setting. The primary motivation of their paper is transfer learning - the embeddings created by their approach are able to generalize to other hospitals and healthcare settings.

Overall I did like this paper. I found it to be easy to read, well motivated, and addressing an important problem in the healthcare domain. As a researcher in this area, it is very true that we are all using our own "siloed" data and do not generally have access to large pre-trained models. I hope that others will produce these kinds of models for the community to use. The authors do not explicitly state that they plan to release their code and pre-trained models, but I sincerely hope that is there intent. If they do not plan to do this, then the impact of this work is dramatically reduced.

However, I do have a few concerns about the paper, listed below:

- It might not be fair to truly call this an unsupervised model. The labels used for evaluation are thresholds on the signals themselves (e.g. SaO2 < 92%) , so the "unsupervised" model actually receives some form of supervision, at least using the current evaluation method. Using a truly different prediction task not directly based on the physiological signals (e.g. mortality, complication during surgery, etc) would provide a cleaner example of unsupervised embeddings that are useful for transfer learning.

- Differences between PHASE and EMA are statistically significant but unlikely to be clinically meaningful - the largest absolute difference in AP is 0.04, and most are much smaller than this. It's unclear if the performance gains enjoyed by PHASE would meaningfully change clinical decision making in any significant way.

- I appreciate the use of XGBoost due to its impressive Kaggle performance, but it strikes me as odd that the authors did not try to fine tune their base model, as that is standard practice for transfer learning. The successes they point to in CV and NLP all use a fine tuning approach, so the evaluation seems incomplete without a performance assessment of fine tuning the base model.

---

> ### Author Response · Authors · 2018-11-27
> **Rebuttal Part 1**
>
>
> We would like to thank the reviewers for their careful consideration of this manuscript and many suggestions for improvement. In response to the reviewers’ comments we have made changes that we feel substantially improve the manuscript and address the reviewers’ concerns, which we have responded to point-by-point.
>
> *"The authors do not explicitly state … reduced."
>
> We thank the reviewer for the excellent point.  We intend to release code pertinent to training the LSTM models, obtaining embeddings, predicting with XGB models, and model stacking feature attributions - submitted as a pull request to the SHAP github (https://github.com/slundberg/shap).  We have indicated our intent to do so in the conclusion (Section 5 Paragraph 3).  Additionally, we intend to release our embedding models, which we recommend for use in forecasting "hypo" predictions.
>
> *"However, I do have a few concerns about the paper, listed below:
> - It might not be fair to truly call this an unsupervised model … useful for transfer learning."
>
> We thank the reviewer for the great point. When we considered prediction problems for our paper, we focused on largely two aspects: (i) clinical importance, and (ii) real-time prediction problems, which are an appropriate evaluation setting for time-series embedding methods. Although predicting mortality makes PHASE a purely unsupervised method, mortality is neither a real-time outcome nor is it reliably measured in our data set. The outcomes we considered - hypoxemia, hypotension, and hypocapnia - are representative adverse real-time events caused by surgery complications and are a significant cause of anesthesia-related complications (Barak et al. Sci. World. Journal, 2015; Curley et al. Crit. Care. Med., 2010). Predicting these events in advance has been considered a promising approach to enable proactive intervention of these events (Lundberg et al. Nature BME 2018).
>
> In order to address the reviewer’s great point, we create a simulated “unsupervised” setting - when predicting each event, we excluded the corresponding physiological signal from our features. For example, we assumed that SaO2 is not recorded when predicting hypoxemia. Under this setting, we must rely on the remaining signals to predict hypoxemia.  This setting is a more unsupervised evaluation in the sense that our outcome is not derived from a signal we create an embedding for. As our results show (Section 6.3; Figure 9), PHASE’s outperformance is consistent in this setting for hypocapnia and hypotension. For hypoxemia, all representations perform poorly because predicting hypoxemia heavily relies on SaO2, leaving little signal for the remaining features.
>
> Finally, to further address the reviewer’s comments, we have mitigated claims of PHASE being unsupervised and instead called our LSTM models “partially supervised” throughout the entirety of the manuscript.  We denote “partially supervised” to mean LSTMs trained with prediction tasks related to the final downstream prediction.  Furthermore, we have refined the discussion in Sections 4.2.1 and 4.2.2 to emphasize that completely unsupervised LSTMs (e.g., autoencoders) are insufficient for downstream “hypo” predictions, which are clinically important perioperative outcomes.  In fact, on our datasets, we found that closeness in the LSTM prediction tasks to the ultimate downstream prediction tasks is beneficial to performance as well as transference.  In order to change the message of our paper, we have added this to our conclusion as well (Section 5 Paragraph 2).

---

> > ### Author Response · Authors · 2018-11-27
> > **Rebuttal Part 2**
> >
> >
> > *"- Differences between PHASE and EMA are statistically significant but … significant way."
> >
> > We thank the reviewer for bringing up a great point.  We argue that like other prediction problems in ML, the percentage improvement is often more relevant than absolute difference in AP.  We believe that our results showing the improvement of PHASE over other representations are significant for the following reasons:
> >
> > 1.	Our intention was to show that PHASE embeddings improve over state-of-the-art prediction models even when using the same dataset. However, the largest clinical impact will likely come from sharing these embeddings, allowing embeddings from a large dataset to be used for a training problem in a smaller dataset. In these situations the AP gain of 0.04 is just a lower bound of the improvement we get from the method, much larger gains are possible when people use these to add power to smaller datasets.
> > 2.	In Lundberg et. al. (2018), the best performing model (analogous to model 4 in Figure 2) is able to achieve higher predictive accuracy than practicing anesthesiologists in predicting hypoxemia and increase doctors’ ability to forecast hypoxemia by providing Shapley value attributions.  With PHASE, we gain further improvement over Prescience (up to 11% improvement in AP - Figure 2f: Model 12 compared to Model 4) and validate a method to obtain Shapley value attributions for our stacked models, leaving little reason to prefer Prescience over PHASE in a hospital.  In health, this level of improvement in predictive performance may impact a number of patients over long periods of time, as pointed out in Lundberg et al. (Nature BME 2018).
> > 3.	The relative improvement of being able to increase precision across all recalls by roughly 2-10% would mean substantially better retrieval of adverse outcomes, beneficial in the face of alarm fatigue and for patient care.  Additionally, the absolute improvement of 0.04 is fairly large given that AP ranges between 0 and 1.
> > 4.	Finally, Model 12 is shown to be significantly better than competing models at a p-value of 0.01 based on one hundred bootstraps of the test set with adjusted pairwise comparisons via ANOVA with Tukey’s HSD test. Moreover, the p-values comparing Model 12 to all others were significant at a much lower threshold than 0.01 (often Model 12 is significantly better even at a threshold of 1e-10).  We have added the p-values for Figures 2 and 3 to the Appendix: Section 6.3 Tables 7 and 8 as well as Section 6.4 Tables 9 and 10.
> >
> > *"- I appreciate the use of XGBoost … of fine tuning the base model."
> >
> > We thank the reviewer for the excellent suggestion.  We have tried fine tuning our LSTM models, with a presentation of the results included in Figures 2 and 3 (MinAtoB, denotes that the best performing LSTM model trained on hospital A data is trained on hospital B data until convergence for each feature).  As one might expect, fine tuned models (14) generally perform on par or better than using just using target hospital data (i.e., without transference) (10).  Additionally, we have modified the discussion in the results, Section 4.2.1 Paragraph 3 to recommend fine tuning as a way to repurpose models in the face of transferring across very different hospitals.

---

### Author Response · Authors · 2018-11-27
**Demarcation of changes for revision**

Most major changes are made in red.  The appendix section is entirely new.

---

### Meta-Review · Area_Chair1 · 2018-12-12
**Method to create unsupervised feature embeddings over physiological signals. Interesting study, but needs additional work.**

**Confidence:** 4
**Recommendation:** Reject

**Metareview:**

Authors present a technique to learn embeddings over physiological signals independently using univariate LSTMs tasked to predict future values. Supervised methods are them employed over these embeddings. Univariate approach is taken to improve transferability across institutions, and Shapley values are used to provide interpretable insight. The work is interesting, and authors have made a good attempt at answering reviewers' concerns, but more work remains to be done.

Pros:
- R1 & R3: Well written.
- R3: Transferrable embeddings are useful in this domain, and not often researched.

Cons:
- R3: Method builds embeddings that assume that future task will be relevant to drops in signals. Authors confirm.
- R3: Performance improvement is marginal versus baselines. Authors essentially confirm that the small improvement is the accurate number.
- R2 & R3: Interpretability evaluation is not sufficient. Medical expert should rate interpretability of results. Authors did not include or revise according to suggestion.